# Isolation and characterization of *Streptomyces* bacteriophages and *Streptomyces* strains encoding biosynthetic arsenals

**Elizabeth T. Montaño**[1], **Jason F. Nideffer**[1], **Lauren Brumage**[1], **Marcella Erb**[1], **Julia Busch**[2], **Lynley Fernandez**[1], **Alan I. Derman**[1], **John Paul Davis**[1], **Elena Estrada**[1], **Sharon Fu**[1], **Danielle Le**[1], **Aishwarya Vuppala**[1], **Cassidy Tran**[1], **Elaine Luterstein**[1], **Shivani Lakkaraju**[1], **Sriya Panchagnula**[1], **Caroline Ren**[1], **Jennifer Doan**[1], **Sharon Tran**[1], **Jamielyn Soriano**[1], **Yuya Fujita**[1], **Pranathi Gutala**[1], **Quinn Fujii**[1], **Minda Lee**[1], **Anthony Bui**[1], **Carleen Villarreal**[1], **Samuel R. Shing**[1], **Sean Kim**[1], **Danielle Freeman**[1], **Vipula Racha**[1], **Alicia Ho**[1], **Prianka Kumar**[1], **Kian Falah**[1], **Thomas Dawson**[1], **Eray Enustun**[1], **Amy Prichard**[1], **Ana Gomez**[1], **Kanika Khanna**[1], **Shelly A. Wanamaker**[1], **Kit Pogliano**[1], **Joe Pogliano**[1]*

1 Division of Biological Sciences, University of California, San Diego, La Jolla, California, United States of America, 2 Department of Immunology, Duke University, Durham, North Carolina, United Stated of America

* jpogliano@ucsd.edu

**Data Availability Statement:** Nucleotide sequences of the phages in this study have been deposited in the GenBank database with accession numbers: MK460245 (BartholomewSD),

## Abstract

The threat to public health posed by drug-resistant bacteria is rapidly increasing, as some of healthcare's most potent antibiotics are becoming obsolete. Approximately two-thirds of the world's antibiotics are derived from natural products produced by *Streptomyces* encoded biosynthetic gene clusters. Thus, to identify novel gene clusters, we sequenced the genomes of four bioactive *Streptomyces* strains isolated from the soil in San Diego County and used Bacterial Cytological Profiling adapted for agar plate culturing in order to examine the mechanisms of bacterial inhibition exhibited by these strains. In the four strains, we identified 104 biosynthetic gene clusters. Some of these clusters were predicted to produce previously studied antibiotics; however, the known mechanisms of these molecules could not fully account for the antibacterial activity exhibited by the strains, suggesting that novel clusters might encode antibiotics. When assessed for their ability to inhibit the growth of clinically isolated pathogens, three *Streptomyces* strains demonstrated activity against methicillin-resistant *Staphylococcus aureus*. Additionally, due to the utility of bacteriophages for genetically manipulating bacterial strains via transduction, we also isolated four new phages (BartholomewSD, IceWarrior, Shawty, and TrvxScott) against *S. platensis*. A genomic analysis of our phages revealed nearly 200 uncharacterized proteins, including a new site-specific serine integrase that could prove to be a useful genetic tool. Sequence analysis of the *Streptomyces* strains identified CRISPR-Cas systems and specific spacer sequences that allowed us to predict phage host ranges. Ultimately, this study identified *Streptomyces* strains with the potential to produce novel chemical matter as well as integrase-encoding phages that could potentially be used to manipulate these strains.

MH669016 (TrvxScott), MK433266 (Shawty), and MK433259 (IceWarrior). Nucleotide sequences of the Streptomyces isolates in this study have been deposited in the GenBank database with accession numbers: SAMN16131985 (DF), SAMN16131988 (SFW), SAMN16131986 (QF2), SAMN16131987 (JS).

**Funding:** These studies were supported by grants from the National Institute of Health AI113295 GM104556, and GM129245.

**Competing interests:** The authors declare no competing interests exist. KP and JP have an equity interest in Linnaeus Bioscience Incorporated, and receive consulting income from the company. The terms of this arrangement have been reviewed and approved by the University of California, San Diego in accordance with its conflict of interest policies.

## Introduction

Antibiotic discovery is an international priority requiring immediate action [1]. The increasing prevalence of multi-drug resistant (MDR) bacterial pathogens has resulted in an increased use of last-resort antibiotics [1–3]. Microbes that produce natural products are the most prolific source of clinically approved antibiotics [4]. Soil dwelling Actinobacteria, notably *Streptomyces*, account for two-thirds of the antibiotics currently on the market [5–7]. Despite intensive studies, however, the full potential of microbes to produce natural products has not been realized [8]. Genome mining studies have shown that microbes encode many biosynthetic gene clusters (BGCs) that have not yet been characterized [8]. It is widely assumed that many of these clusters produce novel natural products and that further characterization of *Streptomyces* bacteria increases the probability of identifying molecules with unique chemical structures and new mechanisms of action [9].

In addition to identifying *Streptomyces* strains containing potentially novel BGCs, it is necessary to improve on the conventional approaches used in natural product antibiotic discovery. One of the major stumbling blocks in natural product discovery is dereplication since the isolation of bioactive molecules often yields antibiotics that have previously been discovered [10]. We recently developed Bacterial Cytological Profiling (BCP) as a new whole-cell screening technique that can be used to rapidly identify the mechanism of action (MOA) of antibiotics [11–16]. BCP can accurately identify the pathway inhibited by antibacterial compounds present in unfractionated crude organic extracts and can be used to guide the purification of molecules with specific bioactivities [11, 15]. BCP can also be used to screen bacterial strains directly on petri plates to identify and prioritize those strains that produce molecules with desired MOAs [15]. In effect, BCP helps with the problem of dereplication by allowing for the determination of the MOA of antibiotics synthesized by a particular *Streptomyces* strain before labor-intensive antibiotic purification efforts are performed.

Since many BGCs are not expressed under laboratory conditions, genetic methods are often used to augment their expression and facilitate the identification and purification of their products [17]. Sometimes, increased expression can be achieved using techniques such as CRISPR/Cas or plasmid cloning and overexpression [17]. However, there is still an occasional need to move large chromosomal regions from one strain to another via transduction to engineer strains optimally suited for antibiotic production. Transduction requires a phage capable of infecting the strain(s) of interest. Moreover, because phages generally display narrow host ranges [18] and relatively few *Streptomyces* phages have been isolated [19] compared to the large number of studied *Streptomyces* bacteria [20], phages aptly suited for genetic manipulations are not available for the majority of antibiotic producing *Streptomyces* strains isolated. In addition, phage derived enzymes such as recombinases and integrases can also be used to engineer new strains [21–25]. Thus, studying the phages that infect antibiotic-producing *Streptomyces* strains could not only yield new transducing phages but potentially also new genetic tools for strain engineering.

Here we describe the isolation and characterization of *Streptomyces* strains and phages. We used a combination of bioinformatics and BCP to characterize the antibiotic biosynthetic potential of four *Streptomyces* strains that displayed an ability to inhibit Gram-negative and Gram-positive bacterial growth. Additionally, we isolated four new phages and assessed their abilities to infect our *Streptomyces* strains, which contained many CRISPRs. The proteins encoded by the phages were subjected to bioinformatic analyses to identify putative integrases that might be used for genetic manipulations. This work highlights a novel set of gene clusters and *Streptomyces* sp. phages that serve as a starting point for the isolation of potentially novel natural products.

## Results and discussion

### Isolation and antibacterial activities of *Streptomyces* sp.

To identify *Streptomyces* strains containing potentially novel BGCs, we collected 28 unique soil samples from sites across San Diego County. From these samples, we isolated a total of eight bacterial strains based on colony morphology. The genus level classification of the eight isolates was confirmed as *Streptomyces* using the phylogeny of their 16S rRNA sequences as well as data from type strains (Fig 1 and Table 1). Each of the strains isolated in this study were

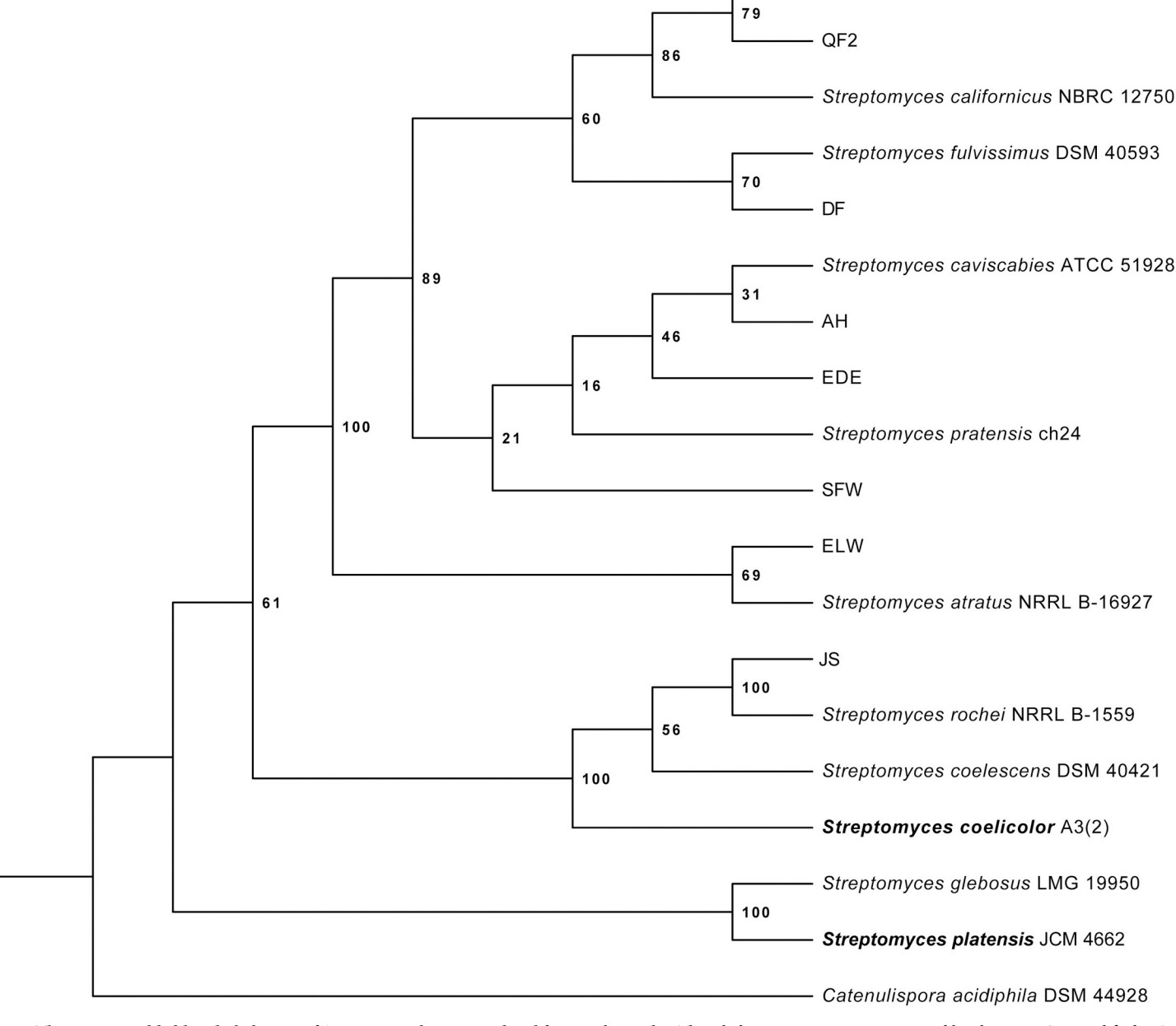

**Fig 1. The maximum likelihood phylogeny of *Streptomyces* bacteria isolated from soil samples.** This phylogenetic tree was constructed by aligning PCR-amplified 16S rRNA sequences with MUSCLE and analyzing with RAxML. Laboratory strains A3(2) and JCM 4662 (in bold) and all type strains are named.

**Table 1. Top NCBI BlastN hits of the 16S rRNA gene sequences.**

| Sample ID | NCBI BlastN Top 16S ribosomal RNA Hit Description | Max Score | Total Score | Query Cover | E value | Percent Identity | Accession No. |
|---|---|---|---|---|---|---|---|
| *S. platensis* JCM 4662 | *Streptomyces platensis* strain JCM 4662 | 2748 | 2748 | 100% | 0 | 100% | NR_024761.1 |
| *S. coelicolor* A3(2) | *Streptomyces coelescens* strain AS 4.1594 | 2793 | 2793 | 98% | 0 | 99.93% | NR_027222.1 |
| JS | *Streptomyces rochei* strain NRRL B-1559 | 2741 | 2741 | 91% | 0 | 99.93% | NR_116078.1 |
| DF | *Streptomyces fulvissimus* strain DSM 40593 | 2691 | 2691 | 100% | 0 | 99.93% | NR_103947.1 |
| QF2 | *Streptomyces californicus* strain NBRC 12750 | 2699 | 2699 | 100% | 0 | 100% | NR_112257.1 |
| EDE | *Streptomyces pratensis* strain ch24 | 1238 | 1238 | 99% | 0 | 98.99% | NR_125616.1 |
| SK | *Streptomyces californicus* strain NBRC 12750 | 1242 | 1242 | 100% | 0 | 100% | NR_112257.1 |
| AH | *Streptomyces pratensis* strain ch24 | 1194 | 1194 | 100% | 0 | 100% | NR_125616.1 |
| ELW | *Streptomyces atratus* strain NRRL B-16927 | 1138 | 1138 | 100% | 0 | 99.84% | NR_043490.1 |
| SFW | *Streptomyces caviscabies* strain ATCC 51928 | 2767 | 2767 | 100% | 0 | 99.87% | NR_114493.1 |

part of a well-supported clade including at least one type strain, These strains (designated JS, DF, QF2, EDE, SK, AH, ELW, and SFW) and two known species (*Streptomyces coelicolor* A3 (2) and *Streptomyces platensis* AB045882) were screened using the cross-streak method for their ability to inhibit the growth of wild type *E. coli* MC4100, an efflux defective mutant *E. coli* JP313 Δ*tolC*, and *B. subtilis* PY79. Since the production of bioactive secondary metabolites is highly dependent on growth conditions, this screen was conducted on actinomycete isolation agar (AIA) as well as Luria Broth (LB) agar. Each of the 10 strains proved capable of inhibiting the growth of *E. coli* and/or *B. subtilis* on at least one of the tested media (Fig 2), suggesting that these strains likely produce antibiotics. As expected, however, the production of antibiotics often depended upon whether the strain was grown on AIA or LB agar. For example, strain ELW was incapable of inhibiting the growth of Gram-negative and Gram-positive bacteria when grown on AIA. However, when grown on LB agar, strain ELW inhibited the growth of both Gram-negative and Gram-positive bacteria. Conversely, strains JS and QF2 exhibited growth inhibition regardless of the media on which they were grown.

## Mechanistic analysis of natural products produced by four *Streptomyces* isolates

Strains QF2, JS, SFW and DF all inhibited the growth of *E. coli* Δ*tolC* when grown on either AIA or LB agar, but in each case, the mechanism underlying inhibition was unknown. Thus, we utilized BCP to examine the mechanism of growth inhibition exhibited by the antibacterial natural products synthesized by these four *Streptomyces* isolates. Each of the four strains was grown on three different media: LB, AIA, or International *Streptomyces* Project-2 media (ISP2) for 5 days to allow for the synthesis and excretion of natural products into the surrounding agar. We then added exponentially growing *E. coli* cells adjacent to the *Streptomyces* lawn. After two hours of incubation at 30°C, the *E. coli* cells were stained with fluorescent dyes and imaged with high resolution fluorescence microscopy. *E. coli* cells incubated adjacent to each of the four *Streptomyces* isolates displayed characteristic cytological profiles that, in some cases, allowed for the classification of these strains according to the MOA of the natural products they produced (Fig 3).

When grown on either LB or ISP2, strain QF2 synthesized an antibiotic that caused the DNA of affected *E. coli* cells to assume a toroidal conformation (Fig 3). This phenotype is characteristic of bacteria treated with protein synthesis inhibitors such as chloramphenicol [11, 26], and thus, we concluded that strain QF2 can synthesize a translation-inhibiting natural product. QF2 also produced a membrane-active secondary metabolite, evidenced by visible

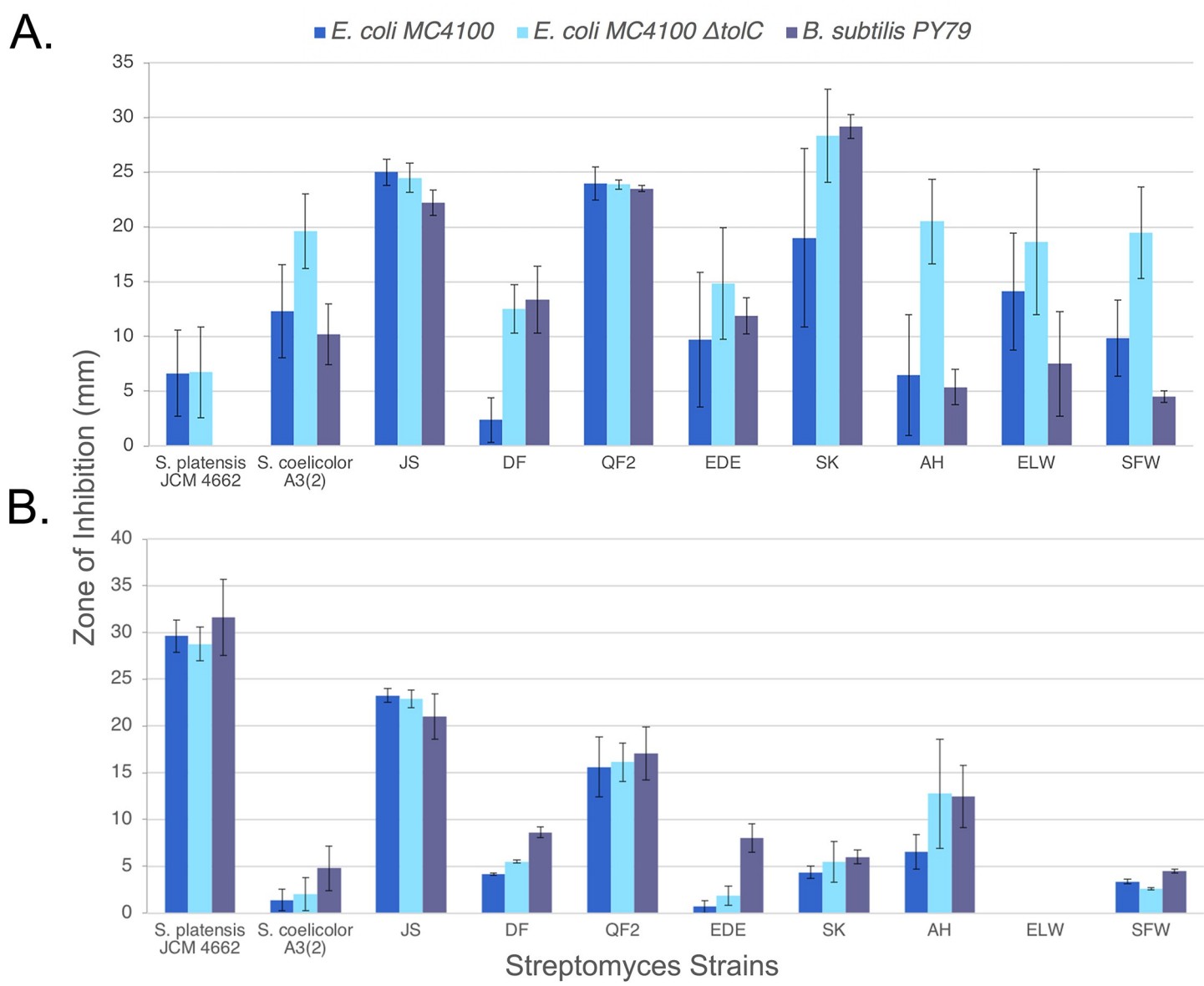

**Fig 2. Inhibition of bacterial growth by *Streptomyces* isolates.** The cross-streak method was used to measure the zone of inhibition among ten *Streptomyces* strains against two Gram-negative *E. coli* strains (MC4100 WT, JP313 Δ*tolC*), and one Gram-positive strain *B. subtilis* PY79 on (**A**) LB and (**B**) AIA. Error bars represent the standard deviation of three independent trials.

membrane abnormalities as well as Sytox Green permeability under all tested nutrient conditions (Fig 3). Strain JS appeared to induce similar phenotypes in *E. coli*, though under different growth conditions; protein synthesis inhibition phenotypes were observed on AIA and ISP2 but not on LB. Similar to strain QF2, Sytox Green permeability was observed in some cells regardless of the medium on which strain JS was grown.

Strain SFW induced distinct phenotypes in *E. coli* cells under each of the three nutrient conditions (Fig 3). On LB, a significant number of *E. coli* cells grown in the presence of strain SFW appeared to contain three chromosomes (white arrows), a phenotype that was not present in the untreated control cells. When strain SFW was grown on AIA, the *E. coli* cells became bent and lost their characteristic rod shape. Finally, strain SFW grown on ISP2 induced

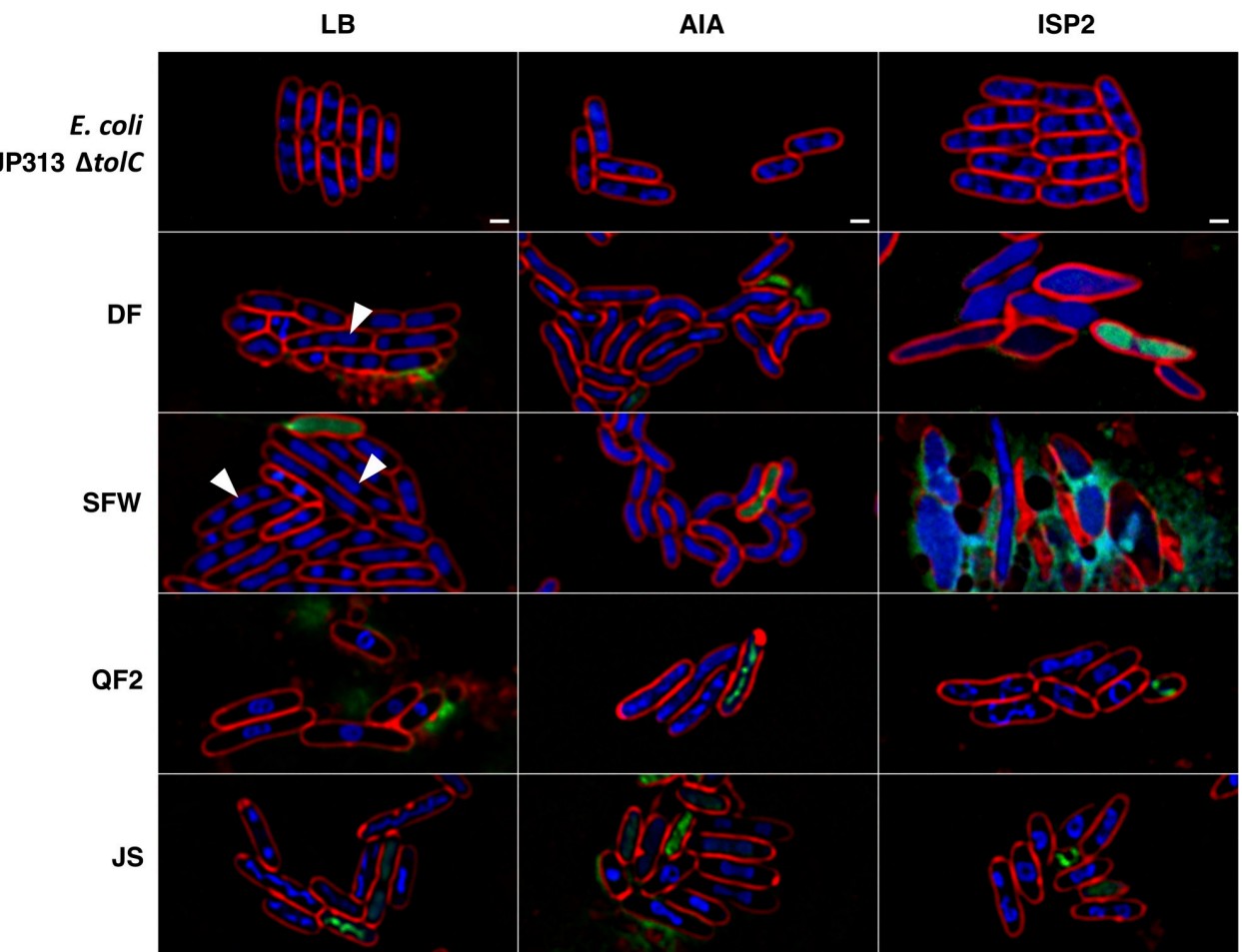

**Fig 3. BCP phenotypes of *E. coli* JP313 Δ*tolC* exposed to natural products produced by four *Streptomyces* soil isolates grown on different solid media.** Also displayed, are *E. coli* JP313 Δ*tolC* untreated controls grown on the tested media (LB agar, AIA, and ISP2 agar). White arrows indicate cells with three chromosomes. BCP images were collected after staining the cells with FM4-64 (red), DAPI (blue), and SYTOX-green (green). The scale bar represents one micron.

substantial swelling in *E. coli* cells that ultimately led to lysis. Notably, *E. coli* cells grown in the presence of strain DF exhibited nearly these same phenotypes under identical growth conditions, suggesting that these two strains produce compounds targeting similar pathways.

## Genomic analysis of four *Streptomyces* isolates

To better understand how strains QF2, JS, SFW and DF inhibited bacterial growth, we sequenced their genomes and aligned them to the most similar genomes in the NCBI database (Fig 4A). Sequence reads for strain DF were assembled into a single contig that was most similar to the genome of *S. fulvissimus* DSM 40593. Sequencing of strains QF2, JS, and SFW yielded multiple contigs that were aligned to the genomes of *S. globisporus* C-1027, *S. parvulus* 12434, and *S. pratensis* ATCC 33331, respectively.

In order to identify predicted gene clusters associated with secondary metabolism, the assembled genome sequences for strains QF2, JS, SFW and DF were annotated using RASTtk [27] and submitted to AntiSMASH 5.0 [28] (Fig 4B). Each strain encoded between 18 and 37 BGCs, some of which were present in multiple strains (Fig 4C). Additionally, some of the

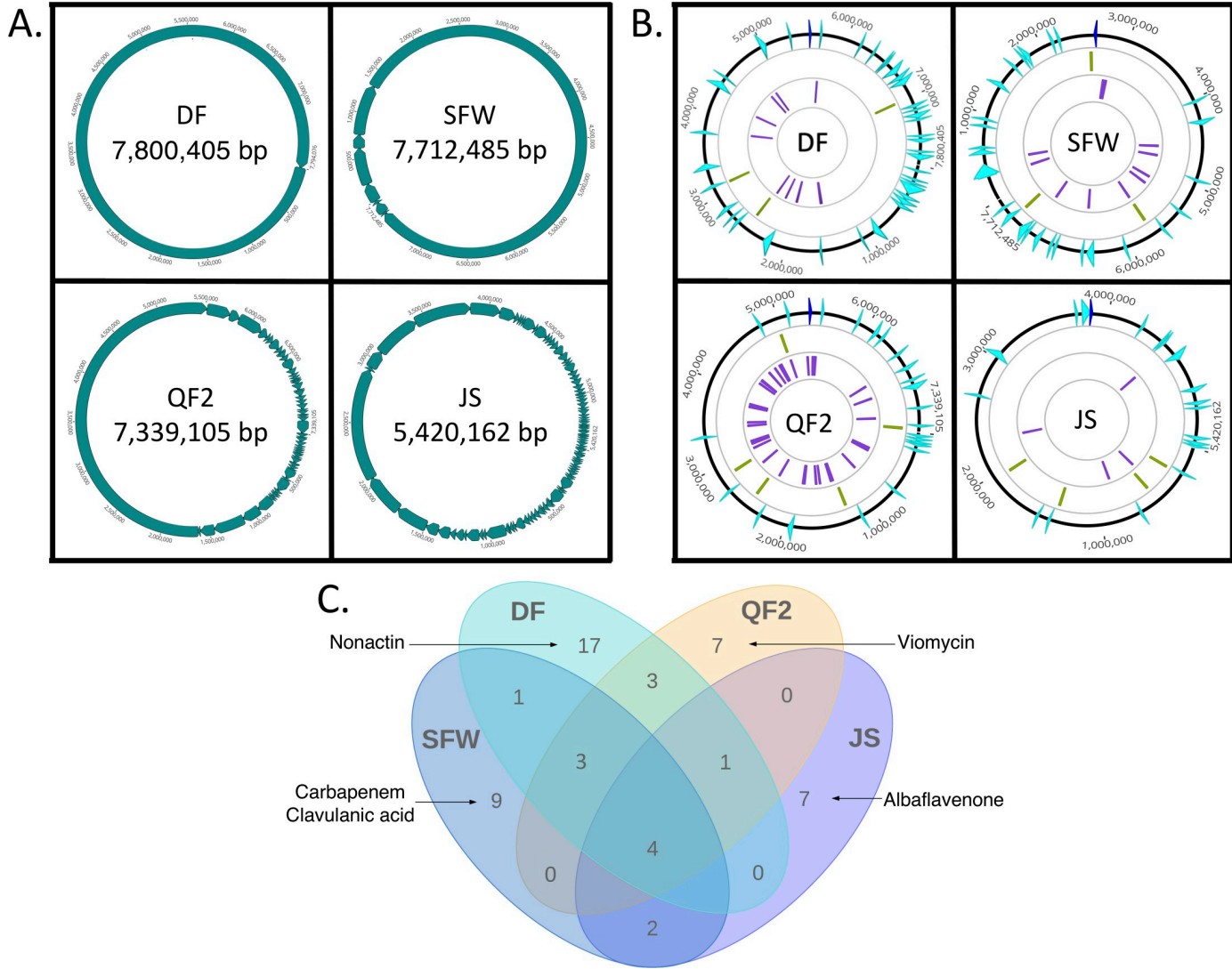

**Fig 4. Genome characteristics of *Streptomyces* strains DF, SFW, QF2, and JS isolated from soil samples.** (**A**) Circularized representations of the linear genomes of the four bacterial isolates displayed as assembled contigs obtained from genome sequencing. (**B**) Genomic annotations are displayed on separate tracks; from outermost to innermost, genomes are oriented according to their threonine operons (dark blue). Predicted biosynthetic gene clusters (light blue), loci of Cas-associated protein-coding genes (green), and CRISPR arrays (purple) are shown. (**C**) A Venn diagram displaying the numbers of BGCs that are shared by and unique to the genomes of our isolates. Five clusters of particular importance are explicitly named.

encoded clusters closely resembled known BGCs in the MIBiG repository [29]. For example, of the 23 putative BGCs identified in the genome of strain QF2 (Table 2), one of them (cluster 21) was similar to the viomycin BGC (Fig 5A). Viomycin inhibits protein synthesis by stabilizing tRNAs in the A site of the bacterial ribosome, inhibiting translocation [30]. According to AntiSMASH, 66% of the genes within the viomycin BGC were similar to genes within cluster 21. However, a global pairwise alignment of cluster 21 and the viomycin BGC revealed that the nucleotide sequence of cluster 21 is actually 98.5% identical over 32.5kb of the 36kb viomycin BGC (Fig 5A). This suggests that a viomycin related molecule is synthesized by strain QF2 and may account for strain's ability to inhibit protein synthesis in *E. coli* (Fig 3). While some of the other clusters in the QF2 genome (Table 2: clusters 2, 4, 7, 8, 9, 12, 13, 22) have significant similarity to known BGCs, no other clusters appear to produce known antibiotics.

**Table 2. BGCs encoded within the draft genome sequence of strain QF2.**

| Strain—QF2 | | | | | | | |
|---|---|---|---|---|---|---|---|
| Cluster | Type | Most Similar MiBIG Cluster and Predicted Percent Similarity | Antibacterial Activity | MIBig BGC-ID | Minimum | Maximum | Length (nt.) |
| 1 | butyrolactone | Coelimycin P1 T1PKS (8%) | | BGC0000038 | 28795 | 39496 | 10702 |
| 2 | terpene | Geosmin Terpene (100%) | | BGC0001181 | 60346 | 82527 | 22182 |
| 3 | NRPS | Griseobactin NRPS (35%) | | BGC0000368 | 101394 | 123548 | 22155 |
| 4 | NRPS | Coelichelin NRPS (72%) | | BGC0000325 | 123549 | 145157 | 21609 |
| 5 | T3PKS | Herboxidiene T1PKS, T3PKS (6%) | | BGC0001065 | 184513 | 202853 | 18341 |
| 6 | terpene | Isorenieratene Terpene (57%) | | BGC0000664 | 681001 | 697369 | 16369 |
| 7 | ectoine | Ectoine Other (75%) | | BGC0000853 | 1125565 | 1134445 | 8881 |
| 8 | T2PKS | Griseorhodin T2PKS (69%) | | BGC0000230 | 1908342 | 1950906 | 42565 |
| 9 | siderophore | Desferrioxamine B Siderophore (80%) | | BGC0000941 | 2278919 | 2290697 | 11779 |
| 10 | LAP,thiopeptide | - | - | - | 2690041 | 2722548 | 32508 |
| 11 | ectoine,butyrolactone | Showdomycin Other (47%) | Nucleic Acid and Protein Synthesis | BGC0001778 | 3344033 | 3359401 | 15369 |
| 12 | melanin | Melanin Other (100%) | | BGC0000911 | 4777563 | 4787988 | 10426 |
| 13 | lanthipeptide | AmfS Lanthipeptide (100%) | | BGC0000496 | 5105643 | 5127766 | 22124 |
| 14 | terpene | - | - | - | 5476141 | 5497007 | 20867 |
| 15 | siderophore | Ficellomycin NRPS (3%) | DNA Replication | BGC0001593 | 5881146 | 5896034 | 14889 |
| 16 | NRPS | Vioprolide A NRPS (25%) | | BGC0001822 | 6093895 | 6137611 | 43717 |
| 17 | bacteriocin | - | - | - | 6194201 | 6205608 | 11408 |
| 18 | NRPS-like | - | - | - | 6430967 | 6442494 | 11528 |
| 19 | NRPS-like,ladderane, arylpolyene | Skyllamycin NRPS (14%) | Unknown MOA | BGC0000429 | 6608620 | 6645073 | 36454 |
| 20 | terpene | Hopene Terpene (46%) | | BGC0000663 | 6755130 | 6764031 | 8902 |
| 21 | NRPS,T1PKS | Viomycin NRPS (66%) | Protein Synthesis | BGC0000458 | 6806327 | 6870853 | 64527 |
| 22 | T3PKS | Alkylresorcinol T3PKS (66%) | Unknown MOA | BGC0000282 | 7058484 | 7080991 | 22508 |
| 23 | lassopeptide | - | - | - | 7260793 | 7282889 | 22097 |

The most similar BGCs in the MIBiG database are listed, as well as the percentage of genes in each MIBiG known cluster that have similarity to genes in the corresponding QF2 cluster. In cases where the most similar known BGC produces an antibiotic, the MOA was listed (Showdomycin [31], Ficellomycin [32], Skyllamycin [33], Viomycin [30], Alkylresorcinol [34]).

Strain JS contained 18 putative BGCs, six of which shared significant similarity (>60% of genes in common) with a known cluster (Table 3). Of these six, however, only cluster 7 was predicted to produce an antibiotic. All of the genes constituting a known terpene cluster that produces albaflavenone were present in cluster 7 (Fig 5B). Albalfavenone is capable of inhibiting the growth of *B. subtilis* by an unknown MOA [35] and has previously been isolated from *S. coelicolor* A3(2) [36], a close relative of strain JS. Since the MOA of albaflavenone is unknown, it's not clear whether the products of cluster 7 or of a different cluster are responsible for the inhibition of protein synthesis and/or the membrane permeability observed in *E. coli* (Fig 3).

Of the 26 putative BGCs that were identified in the genome of strain SFW, only one cluster shared a high percentage of genes in common with a known antibiotic-producing cluster (Table 4). Cluster 1 shared similarity with 62% of the genes within a known BGC that produces carbapenems (Fig 5C), a class of beta-lactam antibiotics that inhibit cell wall biogenesis [41, 42]. Additionally, cluster 4 contained a low percentage of genes in common with a BGC involved in the synthesis of clavulanic acid, which inhibits beta-lactamase and consequently strengthens the bactericidal activity of beta-lactams. Cluster 1 (and perhaps cluster 4) could,

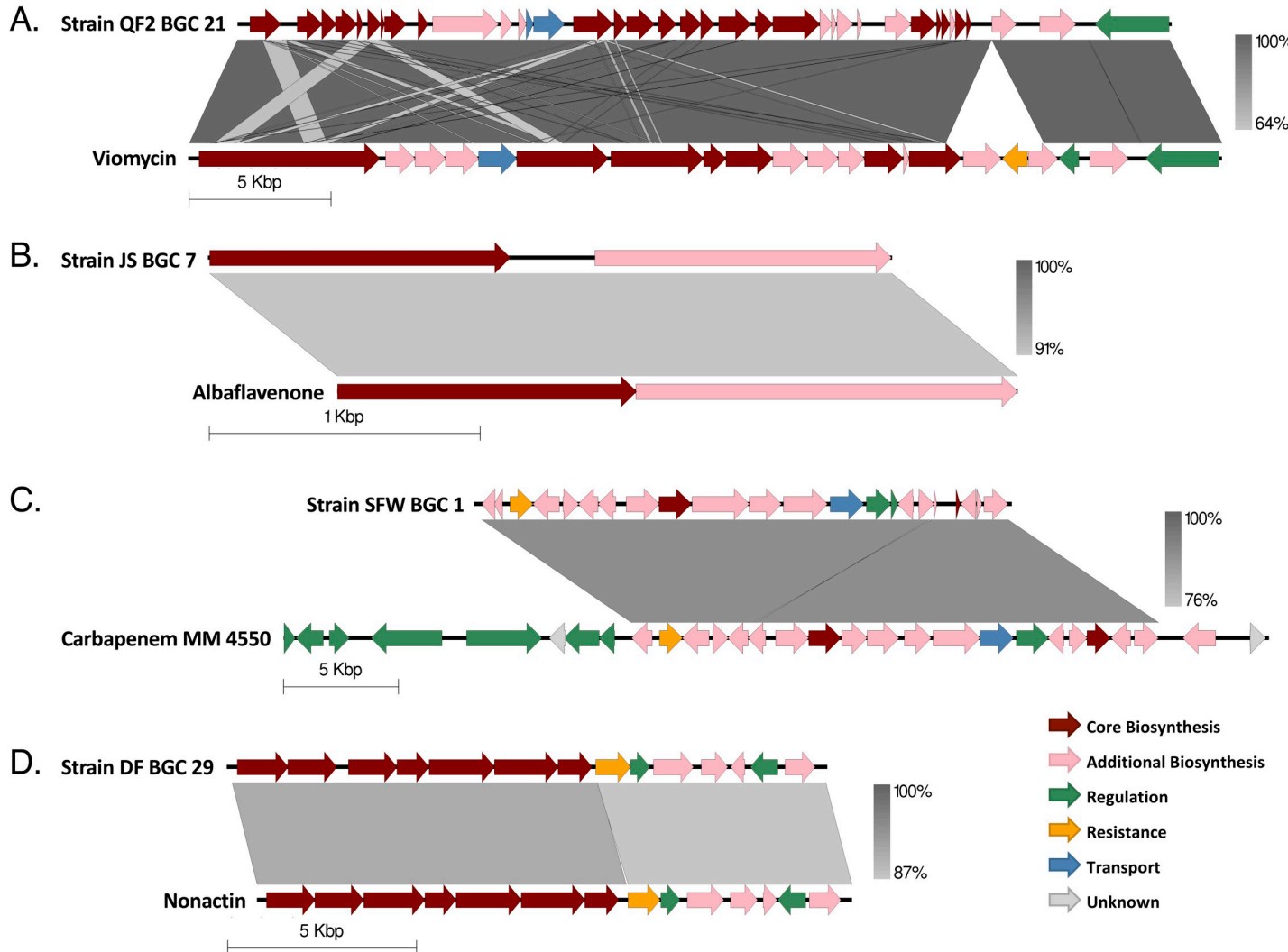

**Fig 5. Comparison of BGCs encoded in the genomes of bacterial soil isolates and the predicted most similar, previously characterized BGC with an antibacterial product.** (**A**) Strain QF2, BGC 21, compared to the BGC previously described to produce the antibiotic viomycin (NCBI Acc No. AY263398.1), encoded in the WGS of *S. vinaceus* ATCC 11861. (**B**) Strain JS, BGC 7, compared to the BGC previously described to produce the antibacterial sesquiterpene Albaflavenone (NCBI Acc No. AL645882.2), encoded in the WGS of *S. coelicolor* A3(2). (**C**) Strain SFW, BGC 1, compared to the BGC previously described to produce the antibacterial beta-lactam Carbapenem MM 4550 (NCBI Acc No. KF042303.1), encoded in the WGS of *S. argenteolus* ATCC 11009. (**D**) Strain DF, BGC 29, compared to the BGC previously described to produce the ammonium ionophore antibiotic Nonactin (NCBI Acc No. AF074603.2), encoded in the WGS of *S. griseus subsp*. *griseus* ETH A7796. Cluster comparisons were constructed in Easyfig. Regions of nucleotide homology are indicated on a gray scale and genes are colored according to the putative function of the corresponding protein product.

therefore, contribute to the synthesis of bioactive molecules that account for the inhibition of *E. coli* cell wall biogenesis on ISP2 media (Fig 3).

Strain DF encoded 37 BGCs (Table 5). Despite this rich supply of BGCs, however, we were only able to identify one cluster that likely participates in the synthesis of an antibiotic with a confirmed MOA. According to AntiSMASH v5.0, cluster 29 shared 92% gene identity with a known BGC that produces nonactin, a bioactive ionophore that disrupts membrane potential [52] (Fig 5D). The known clusters could not fully account for the antibacterial activity exhibited by strain DF (Fig 3), suggesting that antibiotics might be produced by novel clusters.

**Table 3. BGCs encoded within the draft genome sequence of strain JS.**

| Strain—JS | | | | | | | |
|---|---|---|---|---|---|---|---|
| Cluster | Type | Most Similar MiBIG Cluster and Predicted Percent Similarity | Antibacterial Activity | MIBig BGC-ID | Minimum | Maximum | Length (nt.) |
| 1 | T3PKS | Herboxidiene T1PKS, T3PKS (7%) | | BGC0001065 | 214109 | 229085 | 14977 |
| 2 | ectoine | Ectoine Other (100%) | | BGC0000853 | 672467 | 682865 | 10399 |
| 3 | melanin | Melanin Other (60%) | | BGC0000911 | 1414668 | 1425276 | 10609 |
| 4 | siderophore | Desferrioxamine B Siderophore (66%) | | BGC0000941 | 1507497 | 1519344 | 11848 |
| 5 | furan | Methylenomycin Other (9%) | Potentially Inhibits Cell Wall Biosynthesis | BGC0000914 | 2667706 | 2688702 | 20997 |
| 6 | NRPS | Ansamitocin P-3 T1PKS (7%) | | BGC0001511 | 3021294 | 3076234 | 54941 |
| 7 | terpene | Albaflavenone Terpene (100%) | Unknown MOA | BGC0000660 | 3743916 | 3764845 | 20930 |
| 8 | T2PKS | Spore pigment T2PKS (66%) | | BGC0000271 | 3800219 | 3857023 | 56805 |
| 9 | siderophore | - | - | - | 4274698 | 4286154 | 11457 |
| 10 | bacteriocin | - | - | - | 4471015 | 4482384 | 11370 |
| 11 | terpene | - | - | - | 4488880 | 4508472 | 19593 |
| 12 | NRPS | Lipopeptide 8D1-1 & 8D1-2 NRPS (25%) | PMF Collapse | BGC0001370 | 4616241 | 4669719 | 53479 |
| 13 | NRPS | Lipopeptide 8D1-1 & 8D1-2 NRPS (15%) | PMF Collapse | BGC0001370 | 4929292 | 4967094 | 37803 |
| 14 | terpene | Hopene Terpene (76%) | | BGC0000663 | 5042481 | 5069167 | 26687 |
| 15 | terpene | Lysolipin T2PKS (4%) | Cell Wall Biosynthesis | BGC0000242 | 5088579 | 5103665 | 15087 |
| 16 | T1PKS | Candicidin T1PKS (28%) | | BGC0000034 | 5331251 | 5350185 | 18935 |
| 17 | T2PKS, butyrolactone | Kinamycin T2PKS (25%) | DNA Synthesis | BGC0000236 | 5370217 | 5395391 | 25175 |
| 18 | T1PKS | FR-008/Levorin A3 T1PKS (28%) | | BGC0000061 | 5395392 | 5413565 | 18174 |

The most similar BGCs in the MIBiG database are listed, as well as the percentage of genes in each MIBiG known cluster that have similarity to genes in the corresponding JS cluster. In cases where the most similar known BGC produces an antibiotic, the MOA was listed (Methylenomycin [37], Albaflavenone [35], Lipopeptide 8D1-1 & 8D1-2 [38], Lysolipin [39, 40]).

## Antimicrobial activity of four *Streptomyces* isolates against clinically relevant pathogens

To assess the relevance of antibiotics produced by strains JS, DF, SFW, and QF2, we screened their ability to inhibit the growth of three clinically isolated pathogens using the cross-streak method (Table 6). Both strain QF2 and strain JS inhibited the growth of methicillin-resistant *S. aureus* (MRSA) and efflux-deficient *P. aeruginosa* PA01. These strains did not, however, inhibit the growth of the wild-type clinical isolates *P. aeruginosa* PA01 and *P. aeruginosa* P4, which were resistant to the antibiotics produced by all four *Streptomyces* isolates. Strain DF, though incapable of inhibiting the growth of *E. coli tolC*$^+$ (Fig 2), did inhibit the growth of MRSA and efflux-deficient *P. aeruginosa* PA01. Strain SFW was the least capable of inhibiting the growth of clinical pathogens, producing antibiotics only effective against *E. coli tolC*$^+$ (Fig 2).

## Phage isolation and genome sequencing

Phages capable of infecting these newly isolated *Streptomyces* strains can be used for genetic manipulation. With the goal of identifying genetic tools that could be used to augment expression of the BGCs in our bacterial isolates, we isolated bacteriophages using *S. platensis* as a host. This species, in particular, was chosen as a host because it is relatively well-characterized, and *S. platensis* phages capable of infecting our *Streptomyces* isolates could possibly be used to

**Table 4. BGCs encoded within the draft genome sequence of strain SFW.**

| Strain—SFW | | | | | | | |
|---|---|---|---|---|---|---|---|
| Cluster | Type | Most Similar MiBIG Cluster and Predicted Percent Similarity | Antibacterial Activity | MIBig BGC-ID | Minimum | Maximum | Length (nt.) |
| 1 | NRPS, blactam | Carbapenem MM 4550 Other (62%) | Cell Wall Biosynthesis | BGC0000842 | 280243 | 420495 | 140253 |
| 2 | NRPS | Coelichelin NRPS (90%) | | BGC0000325 | 537068 | 587954 | 50887 |
| 3 | terpene | Isorenieratene Terpene (28%) | | BGC0000664 | 601924 | 615201 | 13278 |
| 4 | blactam | Clavulanic acid Other (20%) | Beta-lactamase Inhibition | BGC0000845 | 839861 | 863248 | 23388 |
| 5 | terpene | Hopene Terpene (69%) | | BGC0000663 | 920123 | 946635 | 26513 |
| 6 | T1PKS | Sceliphrolactam T1PKS (72%) | | BGC0001770 | 1325302 | 1388343 | 63042 |
| 7 | bacteriocin | - | - | - | 1598693 | 1609196 | 10504 |
| 8 | lanthipeptide | Kanamycin Saccharide (1%) | Protein Synthesis | BGC0000703 | 1734703 | 1760347 | 25645 |
| 9 | NRPS | Lipopeptide 8D1-1 & 8D1-2 NRPS (6%) | PMF Collapse | BGC0001370 | 1773501 | 1830128 | 56628 |
| 10 | siderophore | Ficellomycin NRPS (3%) | DNA replication | BGC0001593 | 2107372 | 2120491 | 13120 |
| 11 | terpene | - | - | - | 2186197 | 2205874 | 19678 |
| 12 | butyrolactone | Lactonamycin T2PKS (3%) | Protein Synthesis | BGC0000238 | 4018281 | 4029096 | 10816 |
| 13 | NRPS | Istamycin Saccharide (11%) | Protein Synthesis | BGC0000700 | 4251191 | 4307412 | 56222 |
| 14 | siderophore | Desferrioxamine B Siderophore (83%) | | BGC0000941 | 4924798 | 4936579 | 11782 |
| 15 | lanthipeptide | - | - | - | 5321361 | 5346350 | 24990 |
| 16 | terpene | - | - | - | 5588840 | 5608518 | 19679 |
| 17 | ectoine | Ectoine Other (100%) | | BGC0000853 | 6072388 | 6080990 | 8603 |
| 18 | T2PKS, PKS-like | Cinerubin B T2PKS (25%) | DNA Intercalation | BGC0000212 | 6443938 | 6515214 | 71277 |
| 19 | terpene | Steffimycin T2PKS-Saccharide (16%) | | BGC0000273 | 6560271 | 6580717 | 20447 |
| 20 | terpene, ectoine | Ectoine Other (100%) | | BGC0000853 | 6860752 | 6881669 | 20918 |
| 21 | bacteriocin | - | - | - | 6909859 | 6920014 | 10156 |
| 22 | T3PKS | Tetronasin T1PKS (11%) | PMF Collapse | BGC0000163 | 7071589 | 7112647 | 41059 |
| 23 | melanin | Melanin Other (100%) | | BGC0000911 | 7208921 | 7219385 | 10465 |
| 24 | T2PKS, terpene | Spore pigment T2PKS (83%) | | BGC0000271 | 7244899 | 7317424 | 72526 |
| 25 | NRPS | Rimosamide NRPS (21%) | | BGC0001760 | 7458615 | 7511513 | 52899 |
| 26 | butyrolactone | - | - | - | 7592249 | 7602533 | 10285 |

The most similar BGCs in the MIBiG database are listed, as well as the percentage of genes in each MIBiG known cluster that have similarity to genes in the corresponding SFW cluster. In cases where the most similar known BGC produces an antibiotic, the MOA was listed (Carbapenem [42], Clavulanic acid [43], Kanamycin [44], Lipopeptide 8D1-1 & 8D1-2 [38], Ficellomycin [32], Lactonamycin [45, 46], Istamycin [47–49], Cinerubin [50], Tetronasin [51]).

move (via transduction) BGCs from our isolates into a more genetically manipulatable and familiar background [63, 64]. Thus, to increase the probability that our *S. platensis* phages could be used for this purpose, we performed the isolation using the same soil samples from which our *Streptomyces* strains were obtained. Four *S. platensis* actinobacteriophages (BartholomewSD, IceWarrior, Shawty, and TrvxScott) were successfully isolated. These phages were imaged using negative-stain transmission electron microscopy (Fig 6A) and subsequently characterized as members of the family Siphoviridae due to their long filamentous tails and icosahedral capsids [65, 66]. Genome sequencing revealed that BartholomewSD (52.1 kb) and TrvxScott (52.6 kb) are 89% identical (Fig 6B) and belong to the BD2 subcluster of *Streptomyces* phages, which currently contains 20 other members [19]. IceWarrior (55.5 kb) clustered in subcluster BI1 (24 members), and Shawty (40.7 kb) clustered in BB1, a subcluster of 7 phages that includes notable members TG1 and phiC31 (Table 7) [19]. The BLASTp-predicted functions of the gene products encoded by these phages are shown in Table 8.

**Table 5. BGCs encoded within the closed genome sequence of strain DF.**

| Strain—DF Cluster | Type | Most Similar MiBIG Cluster and Predicted Percent Similarity | Antibacterial Activity | MIBig BGC-ID | Minimum | Maximum | Length (nt.) |
|---|---|---|---|---|---|---|---|
| 1 | ectoine | - | - | - | 56852 | 65619 | 8767 |
| 2 | butyrolactone | Coelimycin P1 T1PKS (16%) | | BGC0000038 | 158609 | 168403 | 9794 |
| 3 | terpene | Geosmin Terpene (100%) | | BGC0001181 | 198747 | 220149 | 21402 |
| 4 | transAT-PKS, PKS-like,T1PKS, NRPS | Streptobactin NRPS (76%) | | BGC0000368 | 227951 | 344372 | 116421 |
| 5 | NRPS | Coelichelin NRPS (81%) | | BGC0000325 | 365595 | 413819 | 48224 |
| 6 | NRPS, T1PKS | Arsenopolyketides Other (45%) | Unknown MOA | BGC0001283 | 423679 | 473545 | 49866 |
| 7 | T3PKS | Herboxidiene PKS (6%) | | BGC0001065 | 485172 | 524155 | 38983 |
| 8 | T2PKS | Hiroshidine PKS (41%) | Unknown MOA | BGC0001960 | 862740 | 934422 | 71682 |
| 9 | terpene | Steffimycin D T2PKS-Saccharide (19%) | | BGC0000273 | 1083717 | 1102744 | 19027 |
| 10 | ectoine | Ectoine Other (100%) | | BGC0000853 | 1581580 | 1591978 | 10398 |
| 11 | NRPS, PKS-like | Decaplanin NRPS (7%) | Cell Wall | BGC0001459 | 2187015 | 2263126 | 76111 |
| 12 | lanthipeptide | - | - | - | 2620147 | 2641946 | 21799 |
| 13 | siderophore | Desferrioxamine B Siderophore (100%) | | BGC0000941 | 2701211 | 2711611 | 10400 |
| 14 | NRPS-like | Bottromycin A2 RiPP (39%) | Protein Synthesis | BGC0000469 | 2808481 | 2851819 | 43338 |
| 15 | thiopeptide, LAP | - | - | - | 3106169 | 3139358 | 33189 |
| 16 | NRPS | Phosphonoglycans Saccharide (3%) | | BGC0000806 | 3332988 | 3396052 | 63064 |
| 17 | betalactone | Divergolide A-D T1PKS (6%) | | BGC0001119 | 3744384 | 3772058 | 27674 |
| 18 | T2PKS | Prejadomycin, Rabelomycin, Gaudimycin A, C-D, & UWM6 T2PKS-Saccharide (27%) | Unknown MOA | BGC0000262 | 4307584 | 4380138 | 72554 |
| 19 | lassopeptide | Keywimycin RiPP (100%) | | BGC0001634 | 4424144 | 4446763 | 22619 |
| 20 | T1PKS | Argimycin PI-II, IV-VI, IX & Nigrifactin T1PKS (29%) | | BGC0001433 | 4982495 | 5043259 | 60764 |
| 21 | lanthipeptide | AmfS Lanthipeptide (100%) | | BGC0000496 | 5322352 | 5345015 | 22663 |
| 22 | terpene | - | - | - | 5682792 | 5697907 | 15115 |
| 23 | siderophore | Ficellomycin NRPS (3%) | DNA replication | BGC0001593 | 6155222 | 6169824 | 14602 |
| 24 | butyrolactone | - | - | - | 6322803 | 6333756 | 10953 |
| 25 | bacteriocin | - | - | - | 6494268 | 6503925 | 9657 |
| 26 | terpene | 2-methylisoborneol Terpene (100%) | | BGC0000658 | 6519618 | 6539195 | 19577 |
| 27 | NRPS | Asukamycin T2PKS (12%) | Unknown MOA | BGC0000187 | 6625892 | 6684302 | 58410 |
| 28 | NRPS-like, arylpolyene | Formicamycins A-M PKS (11%) | Unknown MOA | BGC0001590 | 6729705 | 6772824 | 43119 |
| 29 | NRPS | Nonactin T2PKS (92%) | Dissipates Transmembrane Electric Potential | BGC0000252 | 6780952 | 6844946 | 63994 |
| 30 | terpene | Hopene Terpene (69%) | | BGC0000663 | 7099119 | 7125294 | 26175 |
| 31 | linaridin | Pentostatine & Vidarabine Other (9%) | | BGC0001735 | 7147103 | 7167711 | 20608 |
| 32 | T1PKS, NRPS | SGR PTMs NRPS, T1PKS (100%) | Unknown MOA | BGC0001043 | 7205942 | 7253575 | 47633 |
| 33 | bacteriocin | - | - | - | 7266691 | 7277488 | 10797 |
| 34 | melanin | Melanin Other (100%) | | BGC0000911 | 7458714 | 7469181 | 10467 |
| 35 | T3PKS | Herboxidiene T1PKS, T3PKS (9%) | | BGC0001065 | 7501628 | 7542680 | 41052 |
| 36 | terpene | Isorenieratene Terpene (100%) | | BGC0000664 | 7633017 | 7658370 | 25353 |
| 37 | NRPS, T1PKS, LAP, thiopeptide | Lactazole Thiopeptide (33%) | | BGC0000606 | 7671732 | 7738578 | 66846 |

The most similar BGCs in the MIBiG database are listed, as well as the percentage of genes in each MIBiG known cluster that have similarity to genes in the corresponding DF cluster. In cases where the most similar known BGC produces an antibiotic, the MOA was listed (Arsenopolyketides [53], Hiroshidine [54], Decaplanin [55], Bottromycin [56, 57], Prejadomycin, Rabelomycin, Gaudimycin A, C-D, & UWM6 [58], Ficellomycin [59], Asukamycin [60], Formicamycins [61], Nonactin [52], SGR PTMs NRPS [62]).

Table 6. **Inhibition of growth of clinically relevant pathogens by *Streptomyces* strains DF, SFW, QF2, and JS.**

| | Gram-Negative Bacteria | | | | | Gram-Positive Bacteria | |
|---|---|---|---|---|---|---|---|
| | *E. coli* | | *P. aeruginosa* | | | *B. subtilis* | MRSA |
| | JP313 Δ*tolC* | MC4100 | PA01 | P4 | PAO1 Δefflux | PY79 | USA 300 TCH1516 |
| DF | + | - | - | - | + | + | + |
| SFW | + | + | - | - | - | + | - |
| QF2 | + | + | - | - | + | + | + |
| JS | + | + | - | - | + | + | + |

Plus signs indicate growth inhibition, while minus signs indicate pathogen growth.

### Characterization of CRISPR elements in the genomes of our *Streptomyces* strains

Prior to testing the ability of our phages to infect the *Streptomyces* isolates, we decided to examine the strains for complete and functional CRISPR/Cas systems. Our reasoning for this was two-fold. First, the presence of acquired spacers and their specific sequences would allow us to make predictions about whether or not our phages can infect our antibiotic-producing strains. Second, it was conceivable that in doing so we might discover a novel CRISPR/Cas-based system. Our bioinformatic analysis identified the presence of Cas enzymes and CRISPR

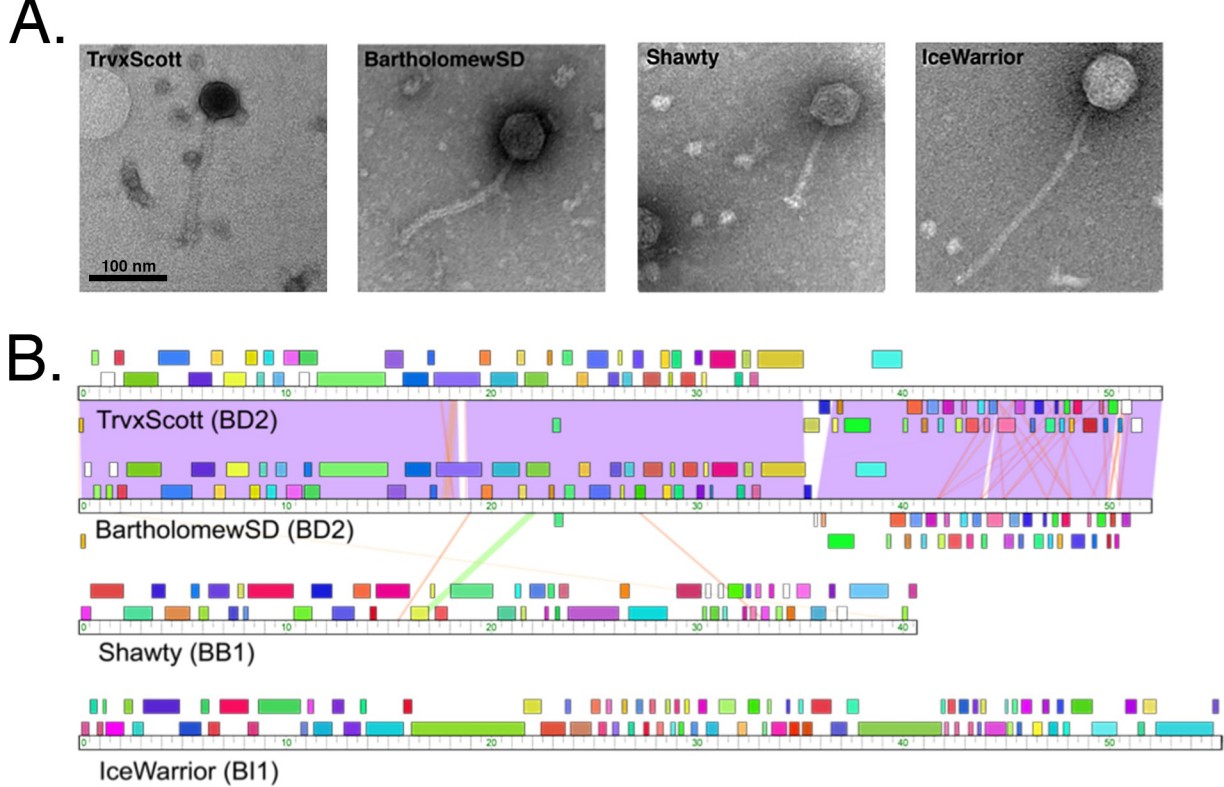

Fig 6. **Characterization of four *Streptomyces* phages isolated from soil samples.** (**A**) Electron micrographs of the four phages (IceWarrior, TrvxScott, BartholomewSD, and Shawty). Lysate samples were negatively stained and imaged with transmission electron microscopy (TEM). The scale bar represents 100 nm. (**B**) A whole-genome sequence comparison of the four phages generated by Phamerator (top to bottom: TrvxScott, BartholomewSD, Shawty, IceWarrior).

**Table 7. Summary of the NCBI WGS annotations of four phage isolates.**

| Bacteriophage | Taxa | Genome (bp) | GC% | Genes | Cluster | Subcluster | Genbank Acc. No. |
|---|---|---|---|---|---|---|---|
| TrvxScott | unclass. Arequatrovirus | 52600 | 67.8 | 81 | BD | BD2 | MH669016 |
| IceWarrior | unclass. Rimavirus | 55532 | 59.5 | 86 | BI | BI1 | MK433259 |
| BartholomewSD | unclass. Arequatrovirus | 52131 | 67.6 | 88 | BD | BD2 | MK460245 |
| Shawty | unclass. Lomovskayavirus | 40733 | 63.2 | 58 | BB | BB1 | MK433266 |

arrays within the genomes all four of our *Streptomyces* isolates, but the abundance of CRISPRs in each strain varied greatly (Table 9). QF2 contained the largest number of predicted CRISPRs– 38 in total, scattered around the chromosome, each containing between one and 25 spacers (Fig 4B, purple; Table 10). Some predicted spacers within these arrays matched with 94–100% identity to sequences within TrvxScott (7 spacers), BartholomewSD (4 spacers), Shawty (2 spacers), and IceWarrior (5 spacers) (Tables 10 and 11). Spacers targeted a variety of genes including those encoding minor tail proteins, tape measure proteins, deoxycytidylate deaminase, helix-turn-helix DNA binding proteins, endolysin, and capsid maturation protease (Fig 7). The large number of putative spacers in the QF2 genome targeting TrvxScott, Bartho-lomewSD, Shawty, and IceWarrior suggests that strain QF2 has likely previously encountered and acquired resistance to each of these phages. Moreover, strain QF2 was isolated from the same soil sample as BartholomewSD, providing support for these findings. Strain QF2 also encoded seven proteins of a Type IE CRISPR-Cas system [67–71]. The QF2 proteins were dis-tantly related to the enzymes of the canonical Cas3 system in *E. coli* (Fig 8), but the operon in strain QF2 lacked two genes (Cas1 and Cas2) involved in spacer acquisition. This phenome-non, the absence of Cas1 and Cas2, has previously been reported as a common feature of Streptomycetaceae Type IE systems [72]. The presence in the QF2 genome of a Cas3 system and spacers mapping to essential proteins in each of the genomes of our phages suggests that the strain is likely resistant to all four of our phages, and thus, transduction is unlikely with strain QF2.

Specific spacers mapping to some of our phages were also discovered within the genomes of strains DF and SFW (but not JS). Strain DF contained two spacers that mapped to sequences within the genome of BartholomewSD, and one of these spacers also shared sequence similar-ity with a region in TrvxScott (Tables 11 and 12). Strain SFW contained two spacers–one that shared sequence similarity with Shawty and another that mapped to a sequence in IceWarrior (Tables 11 and 13). Both strains DF and SFW encoded proteins containing regions with simi-larity to the RuvC and HNH endonuclease domains of known Cas enzymes. However, given the limited similarity of these putative proteins to known Cas proteins, further study is neces-sary to determine if they constitute novel Cas systems. If these systems are functional, we pre-dict that strain DF is resistant to infection by TrvxScott and BartholomewSD, and strain SFW, resistant to Shawty and IceWarrior.

A curious feature emerged from our analysis of the CRISPRs within the *Streptomyces* strains. Among the 205 predicted spacers encoded by all four bacterial strains, 18 contained sequence similarity (95–100% identity) with at least one of the four phages (Table 9). The lengths of the matching sequences (100% identity) within bacterial spacers ranged from 14 to 18 nucleotides and accounted for approximately half the length of a typical spacer. Addition-ally, a single spacer occasionally appeared capable of targeting two distantly related phages. These spacers contained sequences mapping to two distinct genes encoded by different viral genomes. For example, spacer 106 in CRIPSR 23 of strain QF2 is 32 nucleotides in length, and it contains 14 bases that share 100% identity with a region in the TrvxScott tape measure gene. These 14 bases overlap (by 8 nucleotides) with another sequence that is 14 base pairs in length

**Table 8. Functions of the putative proteins encoded within the genomes of four phage isolates.**

| Streptomyces Phage TrvxScott taxon:2301575 | | Streptomyces phage Shawty taxon:2510521 | | Streptomyces phage IceWarrior taxon:2510515 | | Streptomyces phage BartholomewSD taxon:2510587 | |
| --- | --- | --- | --- | --- | --- | --- | --- |
| CDS No. | Product | CDS No. | Product | CDS No. | Product | CDS No. | Product |
| 1 | hypothetical protein | 1 | terminase small subunit | 1 | hypothetical protein | 1 | hypothetical protein |
| 2 | HNH endonuclease | 2 | terminase large subunit | 2 | HNH endonuclease | 2 | hypothetical protein |
| 3 | thioredoxin | 3 | portal protein | 3 | hypothetical protein | 3 | HNH endonuclease |
| 4 | terminase small subunit | 4 | capsid maturation protease | 4 | hypothetical protein | 4 | tRNA-Phe |
| 5 | terminase large subunit | 5 | major capsid protein | 5 | endolysin | 5 | hypothetical protein |
| 6 | portal protein | 6 | head-to-tail adaptor | 6 | head-to-tail connector complex protein | 6 | thioredoxin |
| 7 | capsid maturation protease | 7 | hypothetical protein | 7 | hypothetical protein | 7 | hypothetical protein |
| 8 | scaffolding protein | 8 | major tail protein | 8 | terminase large subunit | 8 | terminase |
| 9 | major capsid protein | 9 | hypothetical protein | 9 | hypothetical protein | 9 | portal protein |
| 10 | head-to-tail connector complex protein | 10 | tail assembly chaperone | 10 | hypothetical protein | 10 | MuF-like minor capsid protein |
| 11 | head-to-tail connector complex protein | 11 | tail assembly chaperone | 11 | hypothetical protein | 11 | scaffolding protein |
| 12 | hypothetical protein | 12 | tape measure protein | 12 | portal protein | 12 | major capsid protein |
| 13 | head-to-tail connector complex protein | 13 | minor tail protein | 13 | hypothetical protein | 13 | head-to-tail adaptor |
| 14 | major tail protein | 14 | minor tail protein | 14 | capsid maturation protease | 14 | head-to-tail stopper |
| 15 | tail assembly chaperone | 15 | minor tail protein | 15 | hypothetical protein | 15 | hypothetical protein |
| 16 | tail assembly chaperone | 16 | minor tail protein | 16 | hypothetical protein | 16 | tail terminator |
| 17 | tape measure protein | 17 | hypothetical protein | 17 | major tail protein | 17 | major tail protein |
| 18 | minor tail protein | 18 | tail fiber | 18 | hypothetical protein | 18 | tail assembly chaperone |
| 19 | minor tail protein | 19 | lysin A | 19 | major tail protein | 19 | tail assembly chaperone |
| 20 | hypothetical protein | 20 | hypothetical protein | 20 | hypothetical protein | 20 | tape measure protein |
| 21 | hypothetical protein | 21 | deoxynucleoside monophosphate kinase | 21 | chitosanase | 21 | minor tail protein |
| 22 | hypothetical protein | 22 | immunity repressor | 22 | hypothetical protein | 22 | minor tail protein |
| 23 | minor tail protein | 23 | Cas4 family exonuclease | 23 | tape measure protein | 23 | hypothetical protein |
| 24 | hypothetical protein | 24 | hypothetical protein | 24 | minor tail protein | 24 | minor tail protein |
| 25 | lysin A | 25 | hypothetical protein | 25 | minor tail protein | 25 | hypothetical protein |
| 26 | hypothetical protein | 26 | hypothetical protein | 26 | hypothetical protein | 26 | hypothetical protein |
| 27 | hypothetical protein | 27 | hypothetical protein | 27 | hypothetical protein | 27 | hypothetical protein |
| 28 | hypothetical protein | 28 | hypothetical protein | 28 | hypothetical protein | 28 | lysin A |
| 29 | hypothetical protein | 29 | hypothetical protein | 29 | holin | 29 | hypothetical protein |
| 30 | exonuclease | 30 | HNH endonuclease | 30 | hypothetical protein | 30 | hypothetical protein |
| 31 | hypothetical protein | 31 | DNA primase | 31 | hypothetical protein | 31 | immunity repressor |
| 32 | hypothetical protein | 32 | restriction endonuclease | 32 | hypothetical protein | 32 | hypothetical protein |
| 33 | hypothetical protein | 33 | DNA polymerase I | 33 | hypothetical protein | 33 | Cas4 family exonuclease |
| 34 | deoxycytidylate deaminase | 34 | RNA polymerase sigma factor | 34 | hypothetical protein | 34 | hypothetical protein |
| 35 | DNA helicase | 35 | hypothetical protein | 35 | hypothetical protein | 35 | hypothetical protein |
| 36 | holliday junction resolvase | 36 | hypothetical protein | 36 | hypothetical protein | 36 | hypothetical protein |
| 37 | hypothetical protein | 37 | hypothetical protein | 37 | hypothetical protein | 37 | deoxycytidylate deaminase |
| 38 | DNA primase | 38 | hypothetical protein | 38 | hypothetical protein | 38 | DnaB-like helicase |
| 39 | DNA primase | 39 | hypothetical protein | 39 | hypothetical protein | 39 | holliday junction resolvase |

*(Continued)*

**Table 8.** (*Continued*)

| CDS No. | Streptomyces Phage TrvxScott taxon:2301575 Product | CDS No. | Streptomyces phage Shawty taxon:2510521 Product | CDS No. | Streptomyces phage IceWarrior taxon:2510515 Product | CDS No. | Streptomyces phage BartholomewSD taxon:2510587 Product |
|---|---|---|---|---|---|---|---|
| 40 | hypothetical protein | 40 | ThyX-like thymidylate synthase | 40 | hypothetical protein | 40 | hypothetical protein |
| 41 | hypothetical protein | 41 | hypothetical protein | 41 | hypothetical protein | 41 | DNA primase |
| 42 | exonuclease | 42 | hypothetical protein | 42 | hypothetical protein | 42 | DNA primase |
| 43 | hypothetical protein | 43 | thioredoxin | 43 | hypothetical protein | 43 | hypothetical protein |
| 44 | HTH DNA binding protein | 44 | hypothetical protein | 44 | hypothetical protein | 44 | hypothetical protein |
| 45 | hypothetical protein | 45 | deoxycytidylate deaminase | 45 | hypothetical protein | 45 | hypothetical protein |
| 46 | ribonucleotide reductase | 46 | hypothetical protein | 46 | hypothetical protein | 46 | Mre11 family dsDNA break repair endo/exonuclease |
| 47 | DNA methylase | 47 | hypothetical protein | 47 | hypothetical protein | 47 | hypothetical protein |
| 48 | hypothetical protein | 48 | hypothetical protein | 48 | hypothetical protein | 48 | helix-turn-helix DNA binding protein |
| 49 | hypothetical protein | 49 | hypothetical protein | 49 | hypothetical protein | 49 | hypothetical protein |
| 50 | hypothetical protein | 50 | hypothetical protein | 50 | hypothetical protein | 50 | ribonucleotide reductase |
| 51 | HTH DNA binding protein | 51 | hypothetical protein | 51 | hypothetical protein | 51 | hypothetical protein |
| 52 | integrase | 52 | hypothetical protein | 52 | hypothetical protein | 52 | hypothetical protein |
| 53 | hypothetical protein | 53 | protein kinase | 53 | hypothetical protein | 53 | hypothetical protein |
| 54 | thymidylate synthase | 54 | integrase | 54 | hypothetical protein | 54 | hypothetical protein |
| 55 | hypothetical protein | 55 | tRNA-Asp | 55 | hypothetical protein | 55 | helix-turn-helix DNA binding protein |
| 56 | hypothetical protein | 56 | tRNA-Thr | 56 | hypothetical protein | 56 | integrase |
| 57 | hypothetical protein | 57 | hypothetical protein | 57 | hypothetical protein | 57 | hypothetical protein |
| 58 | hypothetical protein | 58 | HNH endonuclease | 58 | hypothetical protein | 58 | ThyX-like thymidylate synthase |
| 59 | hypothetical protein | | | 59 | DNA primase/polymerase | 59 | hypothetical protein |
| 60 | hypothetical protein | | | 60 | hypothetical protein | 60 | hypothetical protein |
| 61 | deoxynucleoside monophosphate kinase | | | 61 | hypothetical protein | 61 | hypothetical protein |
| 62 | hypothetical protein | | | 62 | hypothetical protein | 62 | hypothetical protein |
| 63 | hypothetical protein | | | 63 | hypothetical protein | 63 | hypothetical protein |
| 64 | hypothetical protein | | | 64 | hypothetical protein | 64 | hypothetical protein |
| 65 | hypothetical protein | | | 65 | hypothetical protein | 65 | deoxymononucleoside kinase |
| 66 | hypothetical protein | | | 66 | hypothetical protein | 66 | hypothetical protein |
| 67 | hypothetical protein | | | 67 | hypothetical protein | 67 | hypothetical protein |
| 68 | hypothetical protein | | | 68 | hypothetical protein | 68 | hypothetical protein |
| 69 | hypothetical protein | | | 69 | hypothetical protein | 69 | hypothetical protein |
| 70 | hypothetical protein | | | 70 | hypothetical protein | 70 | hypothetical protein |
| 71 | hypothetical protein | | | 71 | hypothetical protein | 71 | hypothetical protein |
| 72 | hypothetical protein | | | 72 | hypothetical protein | 72 | hypothetical protein |
| 73 | hypothetical protein | | | 73 | hypothetical protein | 73 | hypothetical protein |
| 74 | hypothetical protein | | | 74 | hypothetical protein | 74 | hypothetical protein |
| 75 | hypothetical protein | | | 75 | hypothetical protein | 75 | hypothetical protein |
| 76 | hypothetical protein | | | 76 | hypothetical protein | 76 | hypothetical protein |
| 77 | hypothetical protein | | | 77 | hypothetical protein | 77 | hypothetical protein |
| 78 | acetyltransferase | | | 78 | hypothetical protein | 78 | hypothetical protein |
| 79 | hypothetical protein | | | 79 | hypothetical protein | 79 | hypothetical protein |
| 80 | hypothetical protein | | | 80 | hypothetical protein | 80 | hypothetical protein |

(*Continued*)

**Table 8.** (Continued)

| Streptomyces Phage TrvxScott taxon:2301575 | | Streptomyces phage Shawty taxon:2510521 | | Streptomyces phage IceWarrior taxon:2510515 | | Streptomyces phage BartholomewSD taxon:2510587 | |
|---|---|---|---|---|---|---|---|
| CDS No. | Product | CDS No. | Product | CDS No. | Product | CDS No. | Product |
| 81 | hypothetical protein | | | 81 | DNA helicase | 81 | hypothetical protein |
| | | | | 82 | HNH endonuclease | 82 | hypothetical protein |
| | | | | 83 | hydrolase | 83 | hypothetical protein |
| | | | | 84 | DNA helicase | 84 | hypothetical protein |
| | | | | 85 | helix-turn-helix DNA binding domain protein | 85 | hypothetical protein |
| | | | | 86 | hypothetical protein | 86 | hypothetical protein |
| | | | | | | 87 | hypothetical protein |
| | | | | | | 88 | hypothetical protein |

and shares 100% identity with a region within the Shawty genome (Table 11). If these spacers functionally serve to resist infection, our analysis suggests that a single spacer may evolve to efficiently target more than one phage, thus providing broad immunity.

## Susceptibility of *Streptomyces* strains to infection by *S. platensis* phages

With the hope of identifying phages that might serve as tools for transduction, we assessed the susceptibility of our *Streptomyces* isolates to infection by each of the four *S. platensis* phages (Fig 9). As predicted, strain QF2, with its Type IE Cas system and many CRISPRs containing spacers against our phages, could not be infected by any of our four phages. Strain DF experienced inefficient infection by TrvxScott (~2.0 x $10^4$-fold reduced plating efficiency compared to *S. platensis*) and was completely resistant to infection by BartholomewSD. In addition to these results, which were generally predicted by our CRISPR/Cas findings, we also demonstrated strain DF's resistance to infection by Shawty and susceptibility to IceWarrior (~20-fold reduced efficiency). Strain SFW was at least partially resistant to infection by all four phages: Shawty (no infection), BartholomewSD (no infection), IceWarrior (~$10^7$-fold reduced efficiency), and TrvxScott (~1.3 x $10^5$-fold reduced efficiency). Finally, strain JS, despite having no spacers specifically targeting our phages, was similarly immune to infection by Shawty and BartholomewSD and partially resistant to infection by IceWarrior and TrvxScott (~$10^6$-fold and ~$10^5$-fold reduced efficiency, respectively). These data are consistent with our predictions regarding the resistance of our *Streptomyces* isolates to infection by the phages against which they carry spacers, though it is the case that the presence of a spacer did not always confer complete immunity to the phage it targeted. In some cases, strains containing spacers could be

**Table 9. General characteristics of predicted CRISPR-Cas systems within the genomes of strains DF, SFW, QF2, and JS.**

| Strains | CRISPR | Spacers | Repeats | Spacers with Blastn Hits to Host Range Phage | Cas Loci | Cas-Associated Genes |
|---|---|---|---|---|---|---|
| DF | 11 | 13 | 24 | 2 | 3 | 9 |
| SFW | 11 | 23 | 34 | 2 | 3 | 17 |
| QF2 | 38 | 161 | 199 | 14 | 5 | 22 |
| JS | 4 | 8 | 12 | 0 | 4 | 20 |

Included in this table is the number of spacers within the genome of each bacterial strain with sequence similarity to regions within any of the four phage isolates (IceWarrior, TrvxScott, BartholomewSD, or Shawty).

**Table 10. Characteristics of the 38 CRISPRs predicted in the draft genome sequence of strain QF2.**

| Strain QF2 | | | | | | |
|---|---|---|---|---|---|---|
| CRISPRs | Min | Max | Length (nt.) | No. Repeats | No. Spacers | Spacer Blastn Hit to Host Range Phage |
| 1 | 384313 | 384428 | 116 | 2 | 1 | S1 [BartholomewSD] |
| 2 | 437656 | 437755 | 100 | 2 | 1 | |
| 3 | 835820 | 835922 | 103 | 2 | 1 | |
| 4 | 1307659 | 1307752 | 94 | 2 | 1 | |
| 5 | 1344898 | 1345205 | 308 | 6 | 5 | |
| 6 | 1543053 | 1543144 | 92 | 2 | 1 | |
| 7 | 1616159 | 1616542 | 384 | 5 | 4 | S13 [BartholomewSD, TrvxScott] |
| 8 | 1618208 | 1618291 | 84 | 2 | 1 | |
| 9 | 1833873 | 1833977 | 105 | 2 | 1 | |
| 10 | 1861097 | 1861197 | 101 | 2 | 1 | |
| 11 | 2316101 | 2316187 | 87 | 2 | 1 | |
| 12 | 2704894 | 2704990 | 97 | 2 | 1 | |
| 13 | 3015981 | 3016376 | 396 | 7 | 6 | S21 [IceWarrior] |
| 14 | 3106815 | 3107262 | 448 | 8 | 7 | S27 [BartholomewSD, TrvxScott] |
| 15 | 3112452 | 3113433 | 982 | 17 | 16 | S34 [IceWarrior], S41 [TrvxScott] |
| 16 | 3138610 | 3140161 | 1,552 | 26 | 25 | |
| 17 | 3145439 | 3145830 | 392 | 7 | 6 | |
| 18 | 3444685 | 3444779 | 95 | 2 | 1 | |
| 19 | 3507440 | 3507598 | 159 | 3 | 2 | |
| 20 | 3791739 | 3792012 | 274 | 5 | 4 | |
| 21 | 3838730 | 3838804 | 75 | 2 | 1 | |
| 22 | 4257871 | 4258080 | 210 | 4 | 3 | S89 [IceWarrior] |
| 23 | 4327550 | 4328916 | 1,367 | 23 | 22 | S105 [IceWarrior], S106 [Shawty, TrvxScott] |
| 24 | 4333365 | 4334666 | 1,302 | 22 | 21 | S118 [TrvxScott], S131 [Shawty, TrvxScott] |
| 25 | 4335904 | 4336481 | 578 | 10 | 9 | S138 [BartholomewSD], S140 [TrvxScott] |
| 26 | 4522773 | 4522879 | 107 | 2 | 1 | |
| 27 | 4528080 | 4528148 | 69 | 2 | 1 | |
| 28 | 4657265 | 4657374 | 110 | 2 | 1 | |
| 29 | 4754273 | 4754356 | 84 | 2 | 1 | |
| 30 | 4787509 | 4787642 | 134 | 3 | 2 | |
| 31 | 4987714 | 4987810 | 97 | 2 | 1 | |
| 32 | 5305650 | 5305745 | 96 | 2 | 1 | |
| 33 | 5400452 | 5400541 | 90 | 2 | 1 | S151 [IceWarrior] |
| 34 | 5417083 | 5417162 | 80 | 2 | 1 | |
| 35 | 5441923 | 5442032 | 110 | 2 | 1 | |
| 36 | 6552625 | 6552699 | 75 | 2 | 1 | |
| 37 | 6798173 | 6798309 | 137 | 3 | 2 | |
| 38 | 7177566 | 7177797 | 232 | 6 | 5 | |

Spacers with sequence similarity to any of the four phages in this study are listed next to their corresponding CRISPR and are identified according to their position relative to all other spacers within the QF2 genome.

infected relatively inefficiently by the targeted phage. For example, strain DF encoded a single spacer targeting TrvxScott but remained partially susceptible to infection. Strains DF, SFW, and JS were all capable of being infected by TrvxScott and IceWarrior to some degree. Thus, it remains possible that these two phages could be used for transducing BGCs into *S. platensis*.

**Table 11. Spacers within the genomes of strain DF, SFW, and QF2 that have sequence similarity to at least one of the four phage isolates.**

| Strain | CRISPR | Spacer | Blastn Hit to Host Range Phage | Score (bits) / E Val. | No. Identities (%ID) | Strand | Minimum | Maximum | Length (nt.) | Sequence |
|---|---|---|---|---|---|---|---|---|---|---|
| DF | 8 | S10 | BartholomewSD | 30.2 (15) / 2.9 | 15/15 (100%) | Plus / Minus | 4175499 | 4175530 | 32 | TG**CCCACCGGCCGAGCCG**CCTTCCGCAGGCAG |
| | 9 | S11 | TrvxScott | 30.2 (15) / 5.8 | 15/15 (100%) | Plus / Minus | 4689197 | 4689245 | 49 | GGTGTCCCGCCGGTCGCGTGCAT**GTCCTTCGGCTTGA**GCGGGCTGCCG |
| | | | BartholomewSD | 30.2 (14) / 5.8 | 15/15 (100%) | Plus / Minus | | | | |
| SFW | 1 | S1 | Shawty | 28.2 (14) / 5.9 | 14/14 (100%) | Plus / Plus | 250625 | 250650 | 26 | GAGTCACCAGCC**GGGCGAAGGCACGC** |
| | 5 | S6 | IceWarrior | 32.2 (16) / 0.99 | 19/20 (95%) | Plus / Plus | 4553831 | 4553872 | 42 | CGGGGGTCGACGTGACGACGCG**TCGCGTCGTACTTCTCCTTG** |
| QF2 | 1 | S1 | BartholomewSD | 32.2 (16) / 1.2 | 16/16 (100%) | Plus / Plus | 384347 | 384394 | 48 | GCGGACGGCGGC**GGCGGCGGTACCCC**CGGTGTCCACGACGCGGCGGCGG |
| | 7 | S13 | BartholomewSD | 32.2 (16) / 1.9 | 16/16 (100%) | Plus / Plus | 1616356 | 1616423 | 68 | CGACCTGCGGTACCACTCGATCCGGCGGCGGTCCCATCTA**CAAGGGCACGGTCGTC**CAGCGGACCGAG |
| | | | TrvxScott | 32.2 (16) / 1.9 | 16/16 (100%) | Plus / Plus | | | | |
| | 13 | S21 | IceWarrior | 30.2 (15) / 2.5 | 15/15 (100%) | Plus / Minus | 3016077 | 3016109 | 33 | CGCCGGAACCCTCA**AGGAGAGAGAACGGCG**CGGG |
| | 14 | S27 | BartholomewSD | 30.2 (15) / 3.1 | 18/19 (94%) | Plus / Minus | 3106902 | 3106938 | 37 | **AGGGCCTGGCCGTGCGGGG**TGCGGGTGGAGTCGTGGT |
| | | | TrvxScott | 30.2 (15) / 3.1 | 18/19 (94%) | Plus / Minus | | | | |
| | 15 | S34 | IceWarrior | 30.2 (15) / 2.4 | 15/15 (100%) | Plus / Plus | 3112541 | 3112572 | 32 | ACAGCGACGTCGCC**TACAACTACGCCGCC**TGG |
| | | S41 | TrvxScott | 28.2 (14) / 9.5 | 14/14 (100%) | Plus / Plus | 3112961 | 3112992 | 32 | GGTGCTGAACCCGTCG**GCGGCCGTGAACTT**GT |
| | 22 | S89 | IceWarrior | 30.2 (15) / 2.5 | 15/15 (100%) | Plus / Plus | 4257958 | 4257990 | 33 | CCGCGGCGTCTTC**GCCGAGGAGACCCT**GCCC |
| | 23 | S105 | IceWarrior | 30.2 (15) / 2.5 | 15/15 (100%) | Plus / Minus | 4328429 | 4328461 | 33 | CATCAGC**GTCTGAAGCAGCACG**CCCATCGCCTT |
| | | S106 | **Shawty** | 28.2 (14) / 9.5 | 14/14 (100%) | Plus / Plus | 4328491 | 4328522 | 32 | TG*GATCGAGCCGGACGCGGCCACA*TCAGCGGCCC |
| | | | *TrvxScott* | 28.2 (14) / 9.5 | 14/14 (100%) | Plus / Minus | | | | |
| | 24 | S118 | TrvxScott | 28.2 (14) / 9.5 | 14/14 (100%) | Plus / Plus | 4333692 | 4333723 | 32 | GCCGGCGTCCGGC**TACGGCTACGGCT**CCCGCCCC |
| | | S131 | **Shawty** | 28.2 (14) / 9.5 | 14/14 (100%) | Plus / Plus | 4334483 | 4334514 | 32 | AACGCCC*TCCATGAGGCGCCT*G*C*GTTTGGGGTC |
| | | | *TrvxScott* | 28.2 (14) / 9.5 | 14/14 (100%) | Plus / Minus | | | | |
| | 25 | S138 | BartholomewSD | 28.2 (14) / 9.5 | 14/14 (100%) | Plus / Plus | 4336177 | 4336208 | 32 | AACGCGCAGCGATGGCCCGTACGACGGCGCG |
| | | S140 | TrvxScott | 28.2 (14) / 9.5 | 14/14 (100%) | Plus / Minus | 4336299 | 4336330 | 32 | ATCCTCGCCGTCCAG**ACCGCCTCGACGC**AGAT |
| | 33 | S151 | IceWarrior | 36.2 (18) / 0.063 | 18/18 (100%) | Plus / Minus | 5400476 | 5400517 | 42 | GTGGTGGCCTCCCGCACCAGTTC**CTCGGACGCCCTGGGCGG**GCC |

The bold portion of each spacer shares high sequence similarity with a region in the genome of the listed host range phage. In cases where a single spacer mapped to two phages, bold and underlined are used to sequences distinguish the two.

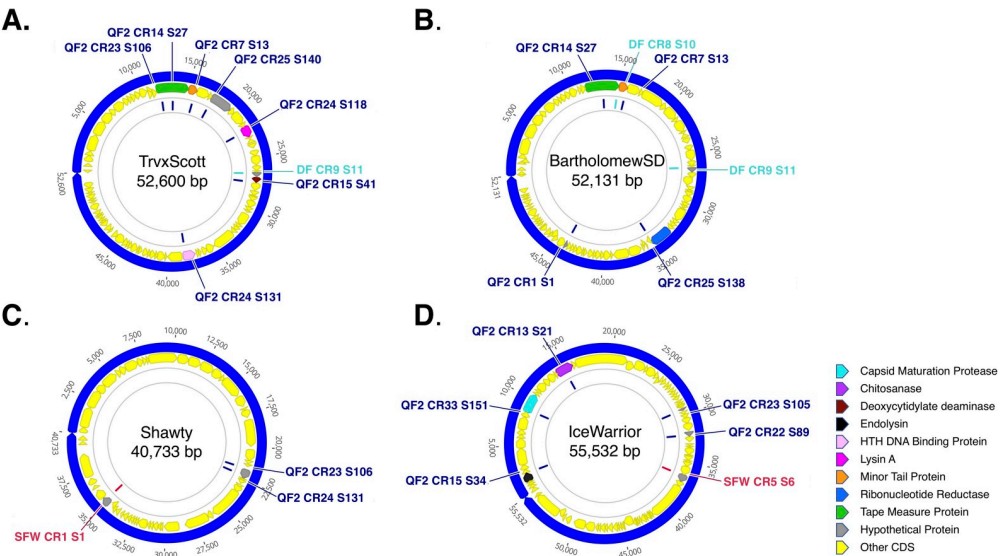

**Fig 7. Genomic maps of phages showing regions containing sequence similarity to spacers found within the CRISPRs of strains QF2, DF, and SFW.** (**A**) TrvxScott, (**B**) BartholomewSD, (**C**) Shawty, and (**D**) IceWarrior. Key displays putative functions of CRISPR targeted genes.

## Identification of phage integrases

In analyzing the proteins encoded within the genomes of our phages, we identified site-specific serine recombinases encoded by BartholomewSD, TrvxScott, and Shawty. The integrases of BartholomewSD and TrvxScott were nearly identical and shared similarity to integrases belonging to a number of previously studied phages. The Shawty integrase shared protein sequence similarity with the integrases of *Streptomyces* phages TG1 and phiC31 (71.3% and 51.9% respectively). The TG1 and phiC31 integrases are distinct recombinases that share 49.7% protein sequence identity and have been used extensively as tools for integrating genes of interest into specific loci within the genomes of a wide variety of organisms, ranging from soil microbes such as *Streptomyces* to multicellular animals such as *Drosophila* [21–25]. Thus,

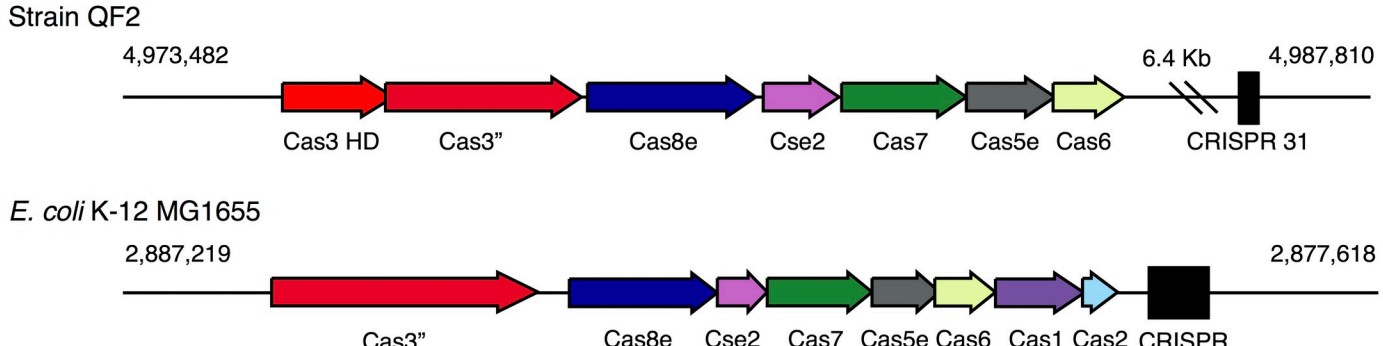

**Fig 8. Class I, Type I-E CRISPR-Cas system encoded in the WGS of strain QF2.** (**top**) The Type I-E CRISPR-Cas operon encoded by strain QG is located from 4,973,482 to 4,987,810 and includes seven genes. The Type I-E cascade is followed by CRISPR 31, consisting of two repeats and a single spacer. (**bottom**) The canonical Type I-E CRISPR-Cas system encoded in the genome of *E. coli* K-12 MG1655 is located from 2,887,219 to 2,877,618 and includes eight genes. The Type I-E cascade is followed by a CRISPR 31, consisting of five repeats and four spacers.

**Table 12. Characteristics of the 11 CRISPRs predicted in the complete genome sequence of strain DF.**

| Strain DF CRISPRs | Min | Max | Length (nt.) | No. Repeats | No. Spacers | Spacer Blastn Hit to Host Range Phage |
|---|---|---|---|---|---|---|
| 1 | 1494904 | 1495000 | 97 | 2 | 1 | |
| 2 | 1520300 | 1520423 | 124 | 2 | 1 | |
| 3 | 1998988 | 1999080 | 93 | 2 | 1 | |
| 4 | 2017315 | 2017508 | 194 | 3 | 2 | |
| 5 | 2245640 | 2245744 | 105 | 2 | 1 | |
| 6 | 2447015 | 2447128 | 114 | 2 | 1 | |
| 7 | 3772076 | 3772180 | 105 | 2 | 1 | |
| 8 | 4175415 | 4175556 | 142 | 3 | 2 | S10 [BartholomewSD] |
| 9 | 4689174 | 4689268 | 95 | 2 | 1 | S11 [BartholomewSD, TrvxScott] |
| 10 | 4806860 | 4806947 | 88 | 2 | 1 | |
| 11 | 5687469 | 5687554 | 86 | 2 | 1 | |

Spacers with sequence similarity to any of the four phages in this study are listed next to their corresponding CRISPR and are identified according to their position relative to all other spacers within the DF genome.

as a newly discovered member of this serine integrase family, the Shawty integrase could also potentially be used to move *Streptomyces*-encoded BGCs between strains to facilitate augmented expression of bioactive natural products.

## Conclusion

This work demonstrates a method for effectively studying newly isolated, antibiotic-producing bacteria and phages that may infect them. We have highlighted how BCP can be used to assess the novelty of BGCs encoded within *Streptomyces* strains, providing another solution to the problem of dereplication. Moreover, our work illustrates how the isolation and genomic analysis of phages that infect antibiotic-producing *Streptomyces* might yield new genetic tools such

**Table 13. Characteristics of the 11 CRISPRs predicted in the draft genome sequence of strain SFW.**

| Strain SFW CRISPRs | Min | Max | Length (nt.) | No. Repeats | No. Spacers | Spacer Blastn Hit to Host Range Phage |
|---|---|---|---|---|---|---|
| 1 | 250601 | 250674 | 74 | 2 | 1 | S1 [Shawty] |
| 2 | 452415 | 452685 | 271 | 3 | 2 | |
| 3 | 2779718 | 2779796 | 79 | 2 | 1 | |
| 4 | 2824197 | 2824275 | 79 | 2 | 1 | |
| 5 | 4553801 | 4554168 | 368 | 5 | 4 | S6 [IceWarrior] |
| 6 | 4731148 | 4731651 | 504 | 10 | 9 | |
| 7 | 5166339 | 5166424 | 86 | 2 | 1 | |
| 8 | 5317268 | 5317371 | 104 | 2 | 1 | |
| 9 | 5722922 | 5723016 | 95 | 2 | 1 | |
| 10 | 6532966 | 6533033 | 68 | 2 | 1 | |
| 11 | 7183989 | 7184080 | 92 | 2 | 1 | |

Spacers with sequence similarity to any of the four phages in this study are listed next to their corresponding CRISPR and are identified according to their position relative to all other spacers within the SFW genome.

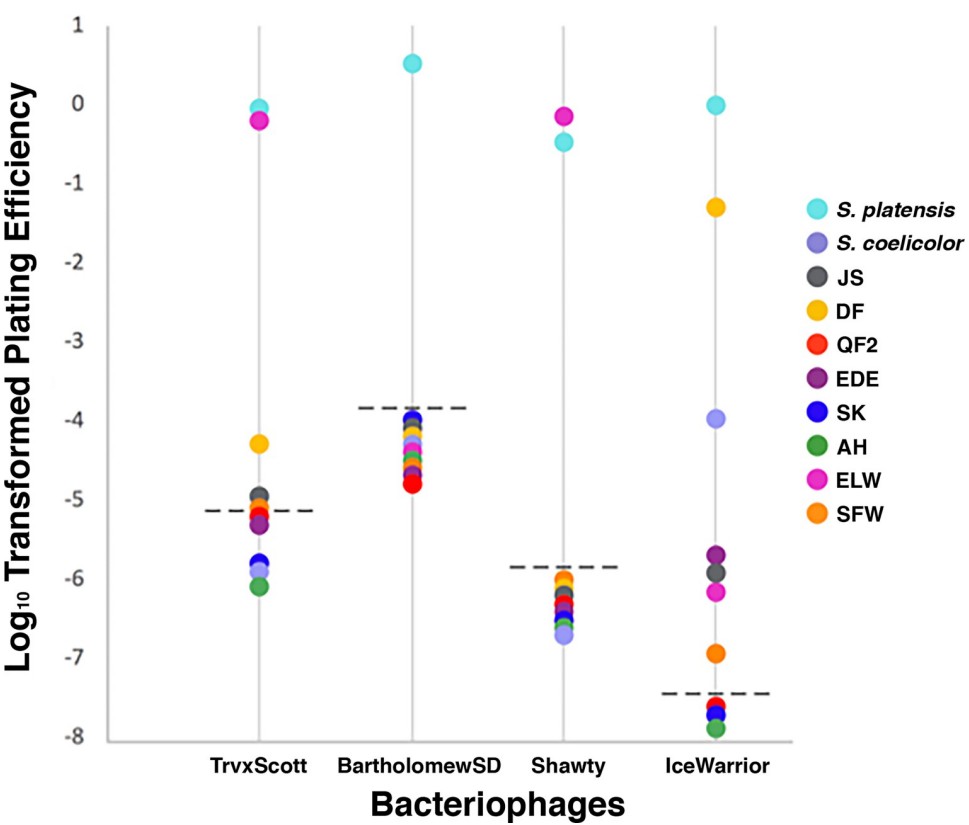

**Fig 9. Host ranges of four phage isolates.** The phages are listed on the horizontal axis, while the vertical axis indicates plating efficiency (log-transformed). Each circle represents one of ten *Streptomyces* bacteria that was tested for susceptibility to phage infection. Circles above the detection limit (dashed line) represent successfully infected strains of *Streptomyces*.

as transducing phages or integrases, which can be used to augment the expression of novel antibiotics. Specifically, strains DF and JS are promising candidates for future novel antibiotic discovery. Each is active against MRSA and has the potential to produce new chemistries given their encoded biosynthetic arsenal and BCP phenotypes. These two strains are also infected by TrvxScott, which might be used to genetically manipulate their BGCs to induce production of novel antibiotics. Ultimately, our identification of potentially novel BGCs and phage integrases serves as a foundation for further studies that could lead to the discovery of new antibiotics.

## Materials and methods

### Soil sample collection and site description

Undergraduate students enrolled in the Phage Hunters Advancing Genomics and Evolutionary Science (PHAGE) class at UCSD collected soil samples for isolating bacteria and their associated phages. Soil samples (approx. 30 ml) were collected around San Diego County (32.7157° N, 117.1611° W), California, USA. GPS coordinates for the phage isolation samples were: Shawty (32.879232 N, 177.237747 W), IceWarrior (32.963038 N, 117.153242 W), TrvxScott (32.882778 N, 117.243333 W), BartholomewSD (32.881200 N, 117.235000 W). Permits were not required for collection because samples were obtained on the UC San Diego campus and students' private property/residences.

## Isolation of *Streptomyces*

Actinomycete isolation agar (AIA) plates (for one liter: sodium caseinate 2 g, L-Asparagine 0.1 g, sodium propionate 4 g, dipotassium phosphate 0.5 g, magnesium sulphate 0.1 g, glycerol 5 mL, rifampicin 50 μgmL⁻¹, nystatin 100 μgmL⁻¹, cycloheximide 100 μgmL⁻¹, agarose 15 g, pH 8.1) were used to select for Actinobacteria from soil samples. 1 g of soil was added to the agar surface and streaked across the AIA plate and incubated for two days at 30˚C. The plates were investigated for individual colonies with morphologies indicative of *Streptomyces* (vegetative hyphae, aerial mycelium), and those colonies were picked and purified at least four times on AIA plates containing 100 μgmL⁻¹ cycloheximide (CHX), which was included to prevent unwanted fungal growth.

## Phage isolation and purification

Actinobacteriophages were isolated from soil samples with host, *Streptomyces platensis* JCM 4664 substrain MJ1A1. An enrichment culture was prepared from 1 g soil and 2.5 ml of *S. platensis* added to 15 ml of Luria-Bertani (LB) medium (for one liter: tryptone 10 g, yeast extract 5 g, NaCl 10 g, agar 15 g, pH 7.0), followed by a 2-day incubation at 30˚C with shaking. Phage were isolated from a 1.2 ml volume of enrichment culture that was centrifuged at maximum speed for 3 min, 1 ml of the resulting supernatant was filtered (0.22 μM filter), and 5 μl of the filtrate was spotted and then streaked onto an LB plate containing 100 μg/ml of cycloheximide. *S. platensis* (0.1 ml) was mixed with 4.5 ml of LB top agar 0.7%, poured over the streak plate and incubated for two days at 30˚C. Resultant plaques were re-streaked onto new LB plates containing 100 μg/ml of cycloheximide about 3–4 times for phage purification.

## Bacterial genomic DNA extraction and quantification for 16S rRNA PCR amplification and sequencing

An adaptation of the DNeasy® Blood & Tissue Kit (Qiagen) protocol was used for bacterial genomic DNA extraction. Strains were cultured overnight at 30˚C in 5 ml of LB broth while rolling. Cells were pelleted (16,000 x g, 3 min) from 1 ml of culture, re-suspended in 180 μl of lysis buffer (prepared in house), and incubated at 37˚C for 45 min after which 25 μl of proteinase K (20 mg/mL) and 200 μl Buffer AL (Qiagen) was added. The samples were vortexed at maximum speed for 20 sec, incubated at 56˚C for 30 min, and 200 μl of ethanol (96–100%) was added. The samples were vortexed at maximum speed for 30 sec, added to a DNeasy Mini spin column, centrifuged (16,000 x g, 1 min), and the supernatant was discarded. Buffer AW2 (Qiagen) was added (500 μl), followed by centrifugation (20,000 x g, 3 min). The DNeasy Mini spin column was placed into a sterile 2 ml microcentrifuge tube, and the gDNA was eluted in 100 μl of AE Buffer by centrifugation (20,000 x g, 1 min) following a 1 min incubation at room temperature. The gDNA concentration was quantified (1 μl sample volume) with a Thermo Scientific™ Nano-Drop™ One Microvolume UV-Vis Spectrophotometer (840274100) and stored at -20˚C.

## Bacterial genomic DNA extraction for PacBio whole-genome sequencing

High molecular weight genomic DNA (20–160 kb) was extracted from four *Streptomyces* strains (DF, SFW, QF2, and JS) with the QIAGEN-Genomic-tip 500/G kit (10262) according to the manufacturer's protocol for bacteria.

## Bacterial whole-genome sequencing, assembly, and annotation

The genome sequences of four *Streptomyces* strains were generated using the Pacific Biosciences RS II (PacBio RS II) single molecule real-time (SMRT) sequencing platform at the IGM

Genomics Center, University of California, San Diego, La Jolla, CA. Linear genome sequences were assembled using the HGAP protocol integrated in the PacBio RS II sequencer (smrt analysis v2.3.0/Patch5) resulting in a variable number (n = 1–95) of contigs per genome, and ranged in size from 5.42 to 7.79 Mb. The mauve contig mover was used to order the contigs of three draft genome sequences (genomes of strains SFW, QF2, and JS) relative to a closely related reference sequence (*S. pratensis* ATCC 33331, *S. globisporus* C-1027, and *S. parvulus* 2297 respectively). DNA sequencing of strain DF resulted in a single contig and did not require reordering to restore gene synteny, however PacBio sequences were combined with Illumina paired end reads. Illumina sequences were generated from a Nextera genomic library and sequenced using the NextSeq 550 platform with the 300 Mbs kit at the Microbial Genome Sequencing Center (MiGS; Pittsburgh, PA). DNA for Illumina sequencing was prepared using the aforementioned protocol for 16S rRNA sequencing. A hybrid assembly of PacBio and Illumina reads was generated in Geneious Prime (2019.2.3) with the following; (1) Illumina paired end reads were processed using the Geneious Prime workflow 'best practice for preprocessing NGS reads in Geneious Prime,' (2) Processed reads were mapped to the PacBio genome using the Geneious assembler with default settings, (3) the resulting consensus sequence was exported (.fasta) for downstream analyses. Gene prediction and annotation were made with the Rapid Annotations using Subsystems Technology (RASTtk) platform [73].

## Phage genomic DNA extraction

5 μl of RNase A and 5 μl of DNase I were added to 10ml of lysate, incubated at 30˚C for 30 minutes, and then precipitated overnight at 4˚C by the addition of 4 ml of 20% polyethylene glycol 8,000. Samples were centrifuged at 10,000 g's for 30 minutes, and pellets resuspended in Qiagen PB buffer and DNA isolated using a Qiagen plasmid DNA isolation column as recommended by the manufacturer.

## Phage genome sequencing, assembly, and annotation

Genomic DNA of 4 actinobacteriophages (TrvxScott, BartholomewSD, Shawty, and IceWarrior) was sequenced using the Illumina MiSeq platform at the Pittsburgh Bacteriophage Institute sequencing facility. The genomes were assembled with Newbler and checked for quality with Consed. The whole genome sequences were submitted to GenBank (Acc No. MH669016, MK460245, MK433266, and MK433259). DNA Master was used for annotation, and NCBI BLASTp was used to determine the potential function of gene products. Whole genome sequence comparisons were performed in Phamerator [74].

## 16S rRNA PCR amplification and sequencing

16S ribosomal DNA templates (~1,465 bp) were amplified using Q5 high fidelity PCR (New England Biolabs) with the universal primer set 27F (5′ -AGAGTTTGATCCTGGCTCAG-3′) and 1492R (5′ -GGTTACCTTGTTACGACTT-3′) [75]. Each PCR mixture (50 μl) contained 100 ng of template gDNA, 500 pmol of each primer, and 200 μM dNTPs. PCR thermocycling conditions were as follows: 30 seconds of initial denaturation at 98˚C, 30 cycles of denaturation at 98˚C for 10 seconds, annealing for 15 seconds at 60˚C, extension at 72˚C for 1.5 minutes, and a final extension at 72˚C for 5 minutes then held at 4˚C. PCR products were purified with the oligonucleotide cleanup protocol as described in the Monarch PCR & DNA Cleanup Kit 5 μg user manual (NEB #T1030). Clean PCR products were sequenced using Sanger methods by Eton Biosciences (https://www.etonbio.com/) and trimmed for quality before analysis.

### CRISPR-Cas sequence analysis and predictions

The sequences of all four *Streptomyces* were searched for CRISPR arrays (repeats and spacers) and potentially associated Cas genes using the following software tools; CRISPR-Cas++ [67, 68], CRISPROne [69] CRISPRDetect [70], and CRISPRMiner2 [71].

### Phylogenetic analyses of bacterial isolates

16S rRNA sequences were trimmed on both ends, (5' and 3') in Geneious Prime using the Trim Ends function with an error probability limit set at 0.05, which trims regions with more than a 5% chance of an error per base. Sequences were aligned using MUSCLE v3.8.425 with a maximum of 1,000 iterations, then maximum likelihood was performed using RAxML with 100 rapid bootstrap replicates and the GTR+G model. The tree was visualized using FigTree v1.4.2.

### Cross-streak method for assessing antibacterial production potential

From a single colony, using sterile Q-tips, *Streptomyces* isolates were streaked in a broad vertical line (2 inch) onto LB, and AIA, solid agar plates and incubated for one week at 30˚C. The day before the assay, test strains (*E. coli* JP313 Δ*tolC*, *B. subtilis* PY79, and *E. coli* MC4100) were grown in 5 ml of LB and incubated at 30˚C overnight while rolling. On the day of the antibacterial screen, the overnight cultures of each test strain were diluted (1:100 in 5 ml LB) and grown to log phase $OD_{600}$ 0.15–0.2 (~1.5 hr at 30˚C while rolling). A volume of 10 μl of each test strain was spotted in distinct lines almost to the edge of the *Streptomyces* line at a perpendicular angle. The plates were incubated overnight at 30˚C, then investigated for the presence of zones of inhibition which were measured in millimeters.

### Bacterial cytological profiling (BCP) on plates

Fluorescence microscopy and BCP on plates was performed as previously described by Nonejuie et al. [15]. Briefly, *Streptomyces* strains (DF, SFW, QF2, and JS) were streaked in a vertical line down the center of LB, AIA, and ISP2 plates (for one liter: 4.0 g Difco yeast extract, 10.0 g Difco malt extract, 4.0 g dextrose, 20.0 g agar, pH 7.0), incubated for one week at 30˚C. The test strain, *E. coli* JP313 Δ*tolC*, was prepared and spotted as described above in the cross-streak method. Following a 2 hr incubation at 30˚C, a 1.5 x 1.5 cm square (~2.5 $cm^2$) piece of agar containing the *E. coli* test strain was cut and prepared for high resolution fluorescence microscopy. The cut piece of agar was placed on a microscope slide, the *E. coli* cells were stained with fluorescent dyes, a coverslip was placed on top of the stained cells then imaged.

### Host range experiment

The host ranges of 4 phages were determined against the *Streptomyces* strains: QF2, DF, JS, and SFW. The experiment was blinded by assigning phages numbers i-iv and hosts letters A-D. A lawn of *Streptomyces* in LB top agar was poured on LB CHX plates. After the top agar solidified, a grid was drawn on the bottom of the plate, and 5 μl of pre-diluted phage samples ($10^0$ to $10^{-10}$ in phage buffer) were spotted in squares on the grid. Plaques were counted and used to calculate a titer, which was then compared to the titer obtained against *S. platensis* to calculate the efficiency of infection.

## Transmission electron microscopy

10 μl of lysate was applied to a copper grid, stained with 1% uranyl acetate, washed twice with phage buffer, and allowed to dry. Images were collected using a FEI Tecnai Spirit G2 BioTWIN Transmission Electron Microscope equipped with a bottom mount Eagle 4k camera.

## Strains used in this study

We used the following strains: *S. platensis* JCM 4664 substrain MJ1A1, *E. coli* MC4100, *B. subtilis* PY79, *P. aeruginosa* P4, *S. aureus* MRSA USA300 TCH1516 from Texas Children's Hospital (USA300-HOU-MR), *S. coelicolor* A3(2) substrain M146, *E. coli* JP313 *ΔtolC*, as well as two strains generously donated by Prof. Keith Poole at Queens University in Kingston, Canada–*P. aeruginosa* PA01 and *P. aeruginosa* K2733 Δefflux (*ΔMexAB–OprM*, *ΔMexCD–OprJ*, *ΔMexEF–OprN*, *ΔMexXY–OprM*). The Δ*tolC5* mutation is derived from strain EW1b (CGSC #5634), and was introduced into strain JP313 [76] by P1 transduction. JP313 was transduced to tetracycline resistance with a lysate of strain CAG18475 (*metC162*::Tn*10*), and the methionine requirement of the transductants was confirmed. This strain was then transduced to prototrophy with a lysate of EW1b, and these transductants were screened on MacConkey agar for the presence of the Δ*tolC5* mutation. EW1b and CAG18475 were obtained from the *Coli* Genetic Stock Center at Yale University.

## Author Contributions

**Conceptualization:** Elizabeth T. Montaño, Jason F. Nideffer, Amy Prichard, Kit Pogliano, Joe Pogliano.

**Data curation:** Elizabeth T. Montaño, Jason F. Nideffer, Julia Busch, Joe Pogliano.

**Formal analysis:** Elizabeth T. Montaño, Jason F. Nideffer, Joe Pogliano.

**Funding acquisition:** Kit Pogliano, Joe Pogliano.

**Investigation:** Elizabeth T. Montaño, Lauren Brumage, Marcella Erb, Lynley Fernandez, John Paul Davis, Elena Estrada, Sharon Fu, Danielle Le, Aishwarya Vuppala, Cassidy Tran, Elaine Luterstein, Shivani Lakkaraju, Sriya Panchagnula, Caroline Ren, Jennifer Doan, Sharon Tran, Jamielyn Soriano, Yuya Fujita, Pranathi Gutala, Quinn Fujii, Minda Lee, Anthony Bui, Carleen Villarreal, Samuel R. Shing, Sean Kim, Danielle Freeman, Vipula Racha, Alicia Ho, Prianka Kumar, Kian Falah, Thomas Dawson, Ana Gomez, Kanika Khanna, Shelly A. Wanamaker.

**Methodology:** Elizabeth T. Montaño, Marcella Erb, Alan I. Derman, Kanika Khanna, Shelly A. Wanamaker, Kit Pogliano.

**Project administration:** Elizabeth T. Montaño, Marcella Erb, Kit Pogliano, Joe Pogliano.

**Resources:** Kit Pogliano, Joe Pogliano.

**Software:** Elizabeth T. Montaño.

**Supervision:** Elizabeth T. Montaño, Marcella Erb, Kanika Khanna, Shelly A. Wanamaker, Kit Pogliano, Joe Pogliano.

**Validation:** Elizabeth T. Montaño, Jason F. Nideffer.

**Visualization:** Elizabeth T. Montaño, Jason F. Nideffer, Julia Busch.

**Writing – original draft:** Elizabeth T. Montaño, Jason F. Nideffer, Joe Pogliano.

**Writing – review & editing:** Elizabeth T. Montaño, Jason F. Nideffer, Marcella Erb, Julia Busch, Lynley Fernandez, Alan I. Derman, Eray Enustun, Amy Prichard, Kit Pogliano, Joe Pogliano.

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
