## [Decision Letter · Decision Letter 0]

28 Jul 2021

PONE-D-21-10682

Isolation and characterization of Streptomyces bacteriophages and Streptomyces strains encoding biosynthetic arsenals

PLOS ONE

Dear Dr. Pogliano,

Thank you for submitting your manuscript to PLOS ONE. After careful consideration, we feel that it has merit but does not fully meet PLOS ONE’s publication criteria as it currently stands. Therefore, we invite you to submit a revised version of the manuscript that addresses the points raised during the review process.

The reviewers recommend reconsideration of your paper following minor revision. Also, kindly check English, grammar and sentence structure throughout the manuscript before submission of the revised version of the paper to PLOS One.

We look forward to receiving your revised manuscript.

Kind regards,

Vijai Gupta, PhD in Microbiology

Academic Editor

PLOS ONE

“These studies were supported by grants from the National Institute of Health AI113295 GM104556, and GM129245.  KP and JP have an equity interest in Linnaeus Bioscience Incorporated, and receive consulting income from the company. The terms of this arrangement have been reviewed and approved by the University of California, San Diego in accordance with its conflict of interest policies.”

“These studies were supported by grants from the National Institute of Health AI113295 GM104556, and GM129245.  KP and JP have an equity interest in Linnaeus Bioscience Incorporated, and receive consulting income from the company. The terms of this arrangement have been reviewed and approved by the University of California, San Diego in accordance with its conflict of interest policies.”

3. We note that Figures 3 & 6 in your submission contain copyrighted images. All PLOS content is published under the Creative Commons Attribution License (CC BY 4.0), which means that the manuscript, images, and Supporting Information files will be freely available online, and any third party is permitted to access, download, copy, distribute, and use these materials in any way, even commercially, with proper attribution. For more information, see our copyright guidelines: http://journals.plos.org/plosone/s/licenses-and-copyright.

a. You may seek permission from the original copyright holder of Figure(s) [#] to publish the content specifically under the CC BY 4.0 license.

Additional Editor Comments (if provided):

The reviewers recommend reconsideration of your paper following minor revision. Also, kindly check English, grammar and sentence structure throughout the manuscript before submission of the revised version of the paper to PLOS One.

Reviewers' comments:

Reviewer's Responses to Questions

**Comments to the Author**

1. Is the manuscript technically sound, and do the data support the conclusions?

Reviewer #1: Yes

Reviewer #2: Yes

2. Has the statistical analysis been performed appropriately and rigorously? 

Reviewer #1: Yes

Reviewer #2: N/A

3. Have the authors made all data underlying the findings in their manuscript fully available?

Reviewer #1: Yes

Reviewer #2: Yes

4. Is the manuscript presented in an intelligible fashion and written in standard English?

Reviewer #1: Yes

Reviewer #2: Yes

5. Review Comments to the Author

Reviewer #1: In the paper entitled “Isolation and characterization of Streptomyces bacteriophages and Streptomyces

strains encoding biosynthetic arsenals”, Pogliano and co-workers describe the isolation of 8 novel strains of Streptomyces, identification of strains that likely produce antibiotics, and proposals of their mechanisms of action using bacterial cytological profiling. Addtionally, 4 novel actinobacteriaphage were characterized

Major concerns:

It is generally quite challenging to determine the species of Streptomyces based solely on the 16s rRNA sequencing. Typically, it is better practice to sequence several housekeeping genes (see Antonie van Leeuwenhoek volume 110, pages563–583 (2017) for more details). It would be useful to do this to determine the likely species that these new Streptomyces are.

Why was neighbor joining chosen for the phylogenetic tree? It seems that a maximum likelihood tree might make more sense? This is not a huge dataset so a maximum likelihood tree should be doable.

Minor concerns:

Figures 7 is a little blurry/difficult to read.

The transition from talking about antibiotics to bacteriophage is a little sudden. Almost feels like they could be 2 different papers.

Reviewer #2: Montaño et al. manuscript entitled "Isolation and characterization of Streptomyces bacteriophages and Streptomyces strains encoding biosynthetic arsenals", Comments given below:

1. Authors should include the future prospectus or outlooks in the manuscript. In reference to which microbes have better performance.

2. In the manuscript microbes name (as Bionomial nomenclature) should be in italics. Please check Table 1 and in the manuscript.

3. In the manuscript lots of tables and figures with same data. So, authors should keep some tables/figs in supplementary.

6. PLOS authors have the option to publish the peer review history of their article (what does this mean?). If published, this will include your full peer review and any attached files.

Reviewer #1: No

Reviewer #2: **Yes: **Ram Prasad

---

## [Author Response · Author response to Decision Letter 0]

5 Oct 2021

Response to reviewers:

Reviewer #1: In the paper entitled “Isolation and characterization of Streptomyces bacteriophages and Streptomyces strains encoding biosynthetic arsenals”, Pogliano and co-workers describe the isolation of 8 novel strains of Streptomyces, identification of strains that likely produce antibiotics, and proposals of their mechanisms of action using bacterial cytological profiling. Additionally, 4 novel actinobacteriaphage were characterized.

Major concerns:

1. It is generally quite challenging to determine the species of Streptomyces based solely on the 16s rRNA sequencing. Typically, it is better practice to sequence several housekeeping genes (see Antonie van Leeuwenhoek volume 110, pages563–583 (2017) for more details). It would be useful to do this to determine the likely species that these new Streptomyces are.

Response: While we agree about the best practices for generating phylogenetic trees for determining species level taxonomy, here we are only interested in assigning these strains at the genus level. We sequenced the 16S rRNA genes from the eight soil bacterial isolates to confirm the genus of each was indeed Streptomyces. Therefore, we believe the 16S rRNA sequence is sufficient for our purposes. We did change the legend to more accurately describe our intentions of generating a genus level phylogenetic tree.

2. Why was neighbor joining chosen for the phylogenetic tree? It seems that a maximum likelihood tree might make more sense? This is not a huge dataset so a maximum likelihood tree should be doable.

Response: We agree that a maximum likelihood tree is a better representation of the genus level phylogeny among the bacterial isolates and have generated a new tree using maximum likelihood to replace Figure 1. 

Minor concerns:

1. Figure 7 is a little blurry/difficult to read.

Response: We uploaded our figure files to the Preflight Analysis and Conversion Engine (PACE) digital diagnostic tool and have generated a high-quality figure to replace the blurred version.

2. The transition from talking about antibiotics to bacteriophage is a little sudden. Almost feels like they could be 2 different papers.

Response: We have added an additional sentence to clarify the rationale for isolating bacteriophage capable of infecting these Streptomyces strains for the purpose of genetically manipulating BGCs for natural product discovery.

Reviewer #2: Montaño et al. manuscript entitled "Isolation and characterization of Streptomyces bacteriophages and Streptomyces strains encoding biosynthetic arsenals", Comments given below:

1. Authors should include the future prospectus or outlooks in the manuscript. In reference to which microbes have better performance.

Response: We have included a section in the conclusions section to include future prospects, specifically noting the microbes of greatest potential for antibiotic discovery. 

2. In the manuscript microbes name (as Bionomial nomenclature) should be in italics. Please check Table 1 and in the manuscript.

Response: We have made this correction.

3. In the manuscript lots of tables and figures with same data. So, authors should keep some tables/figs in supplementary.

Response: We agree with the reviewer that there a large number of tables and figures, but since they are the main sources of data, we believe they belong in the main body of the manuscript where the information can be quickly accessed by the reader. While there is generally no redundancy between the tables, we note that there is a small amount of overlap between Figure 2, which shows the extent of killing of lab strains of E. coli and Bacillus by strains DF, SFW, QF2 and JS on two different media, and Table 6, which both reports whether or not these strains are able to kill clinical isolates of Gram-negative (Pseudomonas PA01 and P4) and Gram-positive pathogens (MRSA). The data for the lab strains of E. coli and Bacillus are intentionally included in both tables for reference.

---

## [Editor Report · Decision Letter 1]

22 Dec 2021

Isolation and characterization of Streptomyces bacteriophages and Streptomyces strains encoding biosynthetic arsenals

PONE-D-21-10682R1

Dear Dr. Pogliano,

We’re pleased to inform you that your manuscript has been judged scientifically suitable for publication and will be formally accepted for publication once it meets all outstanding technical requirements.

Kind regards,

Vijai Kumar Gupta, PhD in Microbiology

Academic Editor

PLOS ONE

Additional Editor Comments (optional):

Authors have answered the questions raised by the reviewers and have modified the manuscript as suggested. The manuscript, in its present form, can be accepted for publication.
---

## [Editor Report · Acceptance letter]

10 Jan 2022

PONE-D-21-10682R1 

Isolation and characterization of *Streptomyces* bacteriophages and *Streptomyces* strains encoding biosynthetic arsenals 

Dear Dr. Pogliano:

I'm pleased to inform you that your manuscript has been deemed suitable for publication in PLOS ONE. Congratulations! Your manuscript is now with our production department. 

Kind regards, 

on behalf of

Dr. Vijai Kumar Gupta 

Academic Editor

PLOS ONE